# Identification of thermotolerant non-canonical PAMs for robust one-pot CRISPR-Cas12a detection

Tian Tian [1,6] ✉, Ting Zhang[1,6], Wanting Zhang[1,6], Zhiqiang Qiu[1], Xinyi Guo[1], Yuxin Chen[1], Mei Lin[1], Weiwei Qi[1], Yuting Shen[1], Mengen Hao[1], Hongrui Xiao[1], Bo Xiang[2], Feibiao Pang[3], Jinzhao Song [4] ✉, Baoqing Sun [2] ✉, Meng Cheng[2] ✉ & Xiaoming Zhou [1,5] ✉

The canonical PAM site TTTV (where V = A, G, or C) is widely used in the design of CRISPR-Cas12a systems for both genome editing and diagnostic applications. Although several non-canonical protospacer-adjacent motifs (PAM) have been identified, they generally exhibit weak Cas12a cleavage activity. In this study, we find that increasing the reaction temperature to 45 °C or higher allows the identification of numerous non-canonical PAMs with *trans*-cleavage activity comparable to that of canonical PAMs, while displaying only weak *cis*-cleavage activity. Moreover, we observe that combining these non-canonical PAMs with elevated temperatures significantly enhances the Cas12a system's ability to discriminate highly similar sequences. Based on these findings, we develop a non-canonical PAM-mediated, poikilothermal, one-pot CRISPR-Cas12a detection platform (POP-CRISPR), which demonstrates substantial improvements in sensitivity, specificity, speed, and target adaptability for nucleic acid detection compared to existing methods. These advantages are validated through the reliable detection of clinical samples, including those of *Human papillomavirus* (HPV), *Mycoplasma pneumoniae* (MP), and its drug-resistant strains. Additionally, we show that POP-CRISPR enables rapid, on-site pathogen detection within 20 min, using a fast sample processing protocol and a miniaturized detection device.

The CRISPR-Cas12a system, first reported in 2015[1], emerges as a key gene editing tool alongside Cas9[2–4]. With the discovery of its trans-cleavage activity in 2018, Cas12a rapidly finds applications in gene diagnostics[5–10]. Similar to the Cas9 system, which relies on the NGG protospacer adjacent motif (PAM), where N represents any nucleotide (A,T,C or G)[11], Cas12a recognizes double-stranded DNA through the TTTV protospacer-adjacent motif (PAM), where V denotes A,G or C.[12,13], which is essential for sequence targeting and localization. The PAM requirement is generally considered a limitation for both gene editing and diagnostics[14]. For instance, if nucleotides are randomly distributed across the genome, the likelihood of encountering an NGG site is 6.25%[15], whereas for TTTV, it is only 1.17%[16]. Furthermore, in certain genomic regions—such as those with mutations, insertions, deletions, or epigenetic modifications—TTTV sites may be absent,

---

[1]School of Life Sciences, South China Normal University, Guangzhou, China. [2]The First Affiliated Hospital of Guangzhou Medical University, Guangzhou, China. [3]Hangzhou EzDx Technology Co., Ltd., Hangzhou, Zhejiang, China. [4]Hangzhou Institute of Medicine, Chinese Academy of Sciences, Hangzhou, Zhejiang, China. [5]MOE Key laboratory of Laser Life Science & Guangdong Provincial Key Laboratory of Laser Life Science, School of Optoelectronic Science and Engineering, South China Normal University, Guangzhou, China. [6]These authors contributed equally: Tian Tian, Ting Zhang, Wanting Zhang. ✉e-mail: ttian@m.scnu.edu.cn; songjinzhao@ucas.ac.cn; sunbaoqing@vip.163.com; ChengMeng@ghmu.edu.cn; zhouxm@scnu.edu.cn

further limiting target availability. Recent studies show that some C-rich PAM sites, such as CTTV, TCTV, and TTCV, exhibit suboptimal PAM activity and are applicable in both gene editing and diagnostics[13,17–19]. However, the limited number of available suboptimal PAM sites and their insufficient cleavage efficiency hinder their broader application. Expanding the understanding of non-canonical PAM sites is crucial for extending the range of target detection, yet comprehensive knowledge of these sites remains limited.

In this study, we first conduct a comprehensive screening of all 256 PAM sites (NNNN, where N = A, T, G, or C) at 37 °C to assess their *trans*-cleavage activity. We find that only five non-canonical PAM sites exhibit *trans*-cleavage activity comparable to that of classical PAMs in the presence of a special targeting sequence. Interestingly, when the reaction temperature is increased to 45 °C or higher, both canonical and non-canonical PAMs show a significant enhancement in *trans*-cleavage efficiency. Notably, the *trans*-cleavage activity of non-canonical PAM sites increases even further, with 82 of them reaching or exceeding the activity level of canonical PAMs. We also observe that combining non-canonical PAMs with higher reaction temperatures significantly improves the Cas12a system's specificity for target sequence recognition. In addition, although the Cas12a system demonstrates high *trans*-cleavage activity under these non-canonical PAM conditions at high reaction temperatures, its *cis*-cleavage activity remained weak. Building on these findings, we develop a robust nucleic acid detection platform, termed the poikilothermal one-pot CRISPR-Cas12a diagnostic (POP-CRISPR). This platform achieves at least an order of magnitude greater sensitivity compared to previous suboptimal PAM-based CRISPR-Cas12a assays (sPAMC)[18]. The

improved sensitivity is validated by significantly higher detection rates and signal-to-noise ratios in clinical detection of *human papillomavirus* (HPV) and *Mycoplasma pneumoniae* (MP). The platform also demonstrates high specificity for accurately detecting drug-resistant MP strains with single-base mutations. Additionally, we develop a compact poikilothermal fluorescence detection device and show that POP-CRISPR can detect MP directly from nasopharyngeal swabs on-site. Using an ultra-fast sample processing protocol (just 2 min), the entire process is completed in 20 min with 100% sensitivity.

## Results

### Identification of numerous non-canonical PAM sites with high trans cleavage activity by increasing the reaction temperature

The TTTV (V = A, G, or C) sequence is the canonical PAM of the CRISPR-Cas12a system and is widely used in gene editing and diagnostic applications[2,4,20] (Fig. 1a). To evaluate how different PAM sites influence the *trans*-cleavage activity of the CRISPR-Cas12a system, we performed an in vitro fluorescence assay to assess the impact of 256 PAM variants (NNNN, where N = A, T, G, C) on Cas12a *trans*-cleavage efficiency. The 256 double-stranded DNA target libraries, each with the same length (294 bp) and recognition sequence but differing in their PAM sites, were amplified by inserting the PAM sites into the primers using plasmid as a template (Fig. 1b, and Supplementary Tables 1 and 5). The inserted PAM sites were confirmed by sequencing (Supplementary Fig. 1), and the DNA target libraries were accurately quantified by gel recovery and Qubit dsDNA fluorescence measurement to ensure the uniformity of target concentration. The *trans*-cleavage activity of LbCas12a, mediated by the 256 different PAMs in target DNA, was

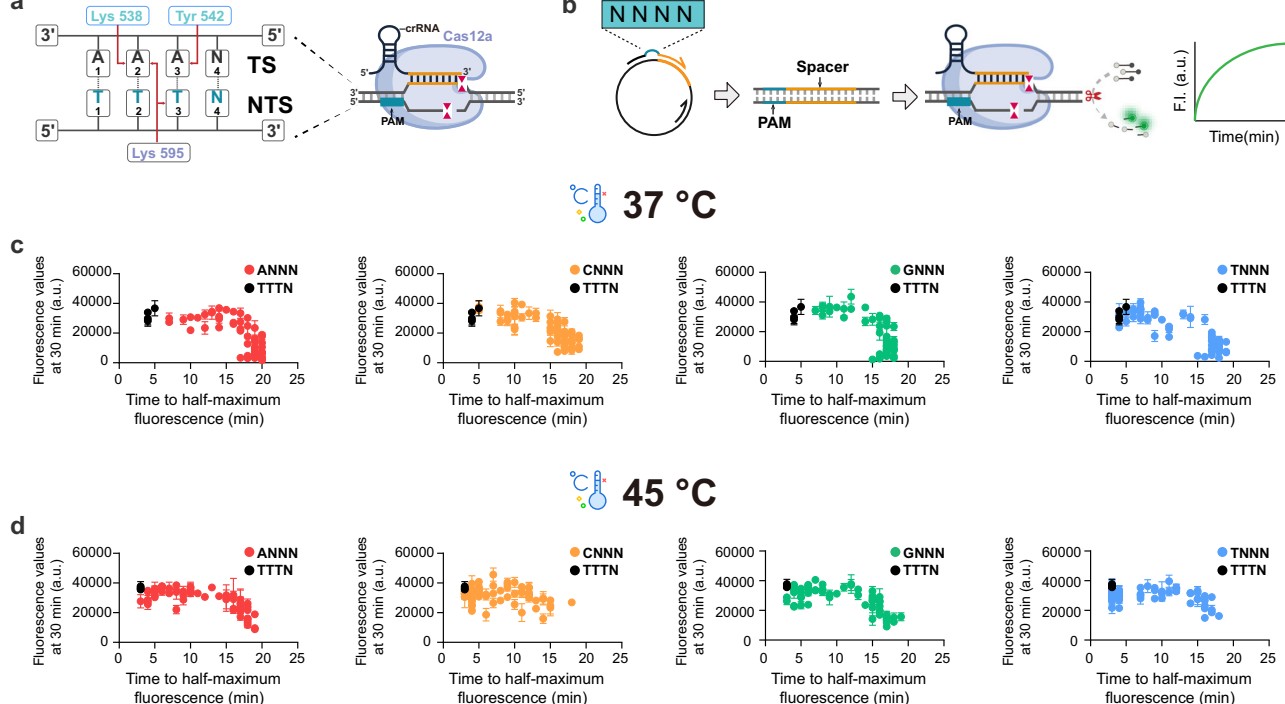

**Fig. 1 | Comprehensive assessment of PAM-mediated collateral activity of LbCas12a at 37 °C and 45 °C. a** Schematic representation of the LbCas12a RNP bound to target double-stranded DNA (dsDNA). The red arrows indicate the specific interactions between the LbCas12a residues and the bases of the PAM site. **b** Construction of a comprehensive PAM target DNA library and evaluation of PAM-mediated collateral activity of Cas12a. Using a circular plasmid as the template, the "NNNN"PAM (N = A/C/G/T) site was inserted into the former primer for PCR, generating a library of 256 target DNA (294 bp) with identical spacer. **c** Summary map of fluorescent kinetics for 256 PAMs in collateral activity experiments at 37 °C. **d** Summary map of fluorescent kinetics for 256 PAMs in collateral activity

experiments at 45 °C. Each dot represents a PAM site, black for canonical PAM, colored for non-canonical PAM (red for ANNN, yellow for CNNN, green for GNNN, blue for TNNN). In these plots, each data point represents a specific PAM site: the *y*-axis denotes the average fluorescence intensity (defined as the maximum fluorescence value) obtained from three replicate reactions at 30 min, while the *x*-axis represents the time required for the fluorescence signal to reach half of that maximum value. The concentrations of dsDNA targets were 1 nM. Data are represented as mean ± standard error (*n* = 3 technical replicates). a.u. represents arbitrary units. Source data are provided as a Source Data file.

initially tested at a DNA concentration of 1 nM at 37 °C, and compared with the classical TTTV sites (Fig. 1c). The experimental results demonstrate that, as previously reported, TTTV mediates the most efficient *trans*-cleavage[20] (Supplementary Data 1). In contrast, the majority of the other PAMs tested exhibited only weak *trans*-cleavage activity (Supplementary Fig. 2a–d). We measured the time ($t$) required to reach half of the maximum fluorescence value at a detection target concentration of 1 nM for each PAM site and used this criterion to classify all 252 non-canonical PAMs into three categories: (1) Best-performing PAM ($t \leq 5$ min); (2) Mediocre-performing PAM (5 min < $t \leq 10$ min); (3) Poor-performing PAM ($t > 10$ min). Based on this classification, we identified only five best-performing PAMs among the tested targets (Supplementary Fig. 3).

Previous studies have shown that higher reaction temperatures enhance detection efficiency under canonical PAM conditions[21,22]. Here, we investigated whether increasing the reaction temperature could influence the *trans*-cleavage activity of LbCas12a for both canonical and non-canonical PAM sites. Interestingly, enhanced trans-cleavage activity of LbCas12a was observed at nearly all PAM sites, with a more significant increase at the non-canonical PAM sites (Fig. 1d and Supplementary Fig. 2e, f). A total of 82 non-canonical PAM sites, which exhibited *trans*-cleavage activity comparable to that of canonical PAM sites and were classified as best-performing PAMs, were identified (Supplementary Fig. 3). We speculate that elevated temperature facilitates local DNA unwinding and enhances the conformational flexibility of the Cas2a complex, thereby lowering the activation barrier for non-canonical PAM recognition and cleavage. In contrast, canonical PAMs possess stronger intrinsic binding affinity and are thus less dependent on temperature elevation. An analysis of the preferential four-base composition of all PAMs revealed that the −3 base plays a pivotal role in Cas12a activity (Supplementary Fig. 4). PAMs with a −3 base of C or T exhibited enhanced *trans*-cleavage activity at 45 °C, while those with a −3 base of A or G showed significantly reduced activity.

### LbCas12a shows its high-temperature tolerance only when it is target-activated

In the previous experiments, we validated the temperature dependence of PAM efficiency by testing the same target sequence. Next, we sought to determine whether this mechanism is universal when the target region is altered. To achieve this, we PCR-amplified and purified a 1000 bp double-stranded DNA segment from the SARS-CoV-2 S gene, identifying 75 canonical PAMs (TTTN) and 585 best-performing non-canonical PAMs. We synthesized 15 crRNAs corresponding to canonical PAMs and 24 crRNAs corresponding to these best-performing non-canonical PAMs (Fig. 2a, b, and Supplementary Tables 2 and 6), and assessed the *trans*-cleavage activity of these PAMs with varying target regions at temperatures of 37, 45, 53, 55 and 57 °C, using target DNA at a final concentration of 100 pM (Fig. 2c). The results showed that the high-temperature-induced improvement in PAM efficiency was consistently observed across different target regions. However, for certain "poor-performing" non-canonical PAMs (such as AAAA, AAAG, CAAT, CAGT, GAAA, TAAA), increasing the temperature had minimal effect on enhancing *trans*-cleavage activity (Supplementary Fig. 5).

Moreover, we found that although nearly all tested sites exhibited the highest *trans*-cleavage activity at 45 °C, the majority of sites displayed comparable activity at 53 °C to that at 45 °C. As the temperature increased further, *trans*-cleavage efficiency at some PAM sites decreased, although several sites still maintained relatively high activity at 55 °C. Notably, certain crRNAs, such as crRNA1, 10, 11, 20, 28, and 30, maintained high *trans*-cleavage activity even at the elevated temperature of 57 °C. We further tested the performance of these high-temperature-effective crRNAs, including crRNA7, 8, 10, and 12, at 60 °C, but found that their activity significantly decreased (Supplementary Fig. 6). To better understand this temperature dependence,

we next examined target-type effects. Notably, this high-temperature tolerance mechanism appears to be more applicable to double-stranded DNA. When testing crRNAs with single-stranded targets, we observed that the highest activity was typically achieved at 37 °C (Supplementary Fig. 7). This difference likely arises from distinct activation mechanisms: elevated temperature promotes R-loop formation and activation with dsDNA targets, whereas ssDNA activation depends solely on crRNA–target hybridization[23], which becomes destabilized at higher temperatures. Given this thermal sensitivity, we speculate that crRNA degradation may also contribute to activity loss at high temperatures[24]. Therefore, we selected four crRNAs, which showed a decrease in *trans*-cleavage activity under high-temperature conditions, and performed phosphorothioate modification at both ends to enhance their stability. The results indicated that phosphorothioate modification indeed improved *trans*-cleavage activity at elevated temperatures (Supplementary Fig. 8).

Together, these results suggest that while crRNA engineering can improve high-temperature performance, intrinsic enzyme thermostability remains limiting. Thus, inspired by the observation that enAsCas12a retains its activity at elevated temperatures (60 °C)[25], we next examined whether the thermostability of LbCas12a could be improved by rational mutagenesis. To this end, we introduced three substitutions (D156R/G532R/K538R) in LbCas12a, analogous to the corresponding mutations (E174R/S542R/K548R) in enAsCas12a (Supplementary Fig. 9). However, the LbCas12a-Mut exhibited a temperature-dependent *trans*-cleavage pattern similar to that of the LbCas12a-WT, showing no enhancement in activity above 57 °C (Supplementary Fig. 10a). In contrast, optimizing the reaction buffer (e.g., using Tango buffer) substantially enhanced high-temperature *trans*-cleavage activity (Supplementary Fig. 10b), allowing both LbCas12a-WT and LbCas12a-Mut to remain functional at temperatures where the standard buffer conditions failed (Supplementary Fig. 11). Additionally, we assessed whether other Cas12a proteins, such as AsCas12a and FnCas12a, exhibited similar heat tolerance effects as LbCas12a. It appears that these Cas12a systems share similar temperature dependence, although LbCas12a demonstrated superior heat tolerance (Supplementary Fig. 12).

We then evaluated which step in the LbCas12a reaction system contributes to its heat tolerance. We found that LbCas12a itself is highly heat-sensitive. Pre-incubation of LbCas12a at 37 °C, 45 °C, or 53 °C for 30 min led to a complete loss of activity (Fig. 2d). However, when only the crRNA was exposed to the same temperature conditions, the LbCas12a system remained functional (Fig. 2e). The LbCas12a protein-crRNA complex exhibited greater heat stability, as pre-incubation at 37 °C did not result in any loss of activity, though treatments at 45 °C and 53 °C still caused significant inactivation (Fig. 2f). When LbCas12a, crRNA, and target DNA were all present in the reaction, the Cas12a system exhibited excellent heat tolerance, suggesting that this resistance is primarily associated with the DNA-target-activated state (Fig. 2g). Furthermore, we tested the maintenance of its *trans*-cleavage activity and found that the activated Cas12a protein retained its activity with only minimal reduction after being stored for up to 2 months at 4 °C, 25 °C, and 37 °C. Even at 45 °C, its activity remained stable for nearly a month (Supplementary Fig. 13).

### Cas12a system presents weak cis-cleavage activity but exhibits strong trans-cleavage activity on multiple substrates at high reaction temperatures

The preceding measurements focused on evaluating the *trans*-cleavage efficiency of the Cas12a system. The goal of the subsequent investigation was to determine the effect of elevated temperatures on the *cis*-cleavage behavior of the Cas12a system. Using the same 1000 bp S-gene fragment as the target, we assessed the temperature-dependent *cis*-cleavage activity with crRNAs corresponding to two canonical PAMs (TTTA) (Fig. 3a, b) and two non-canonical PAMs (TCCA

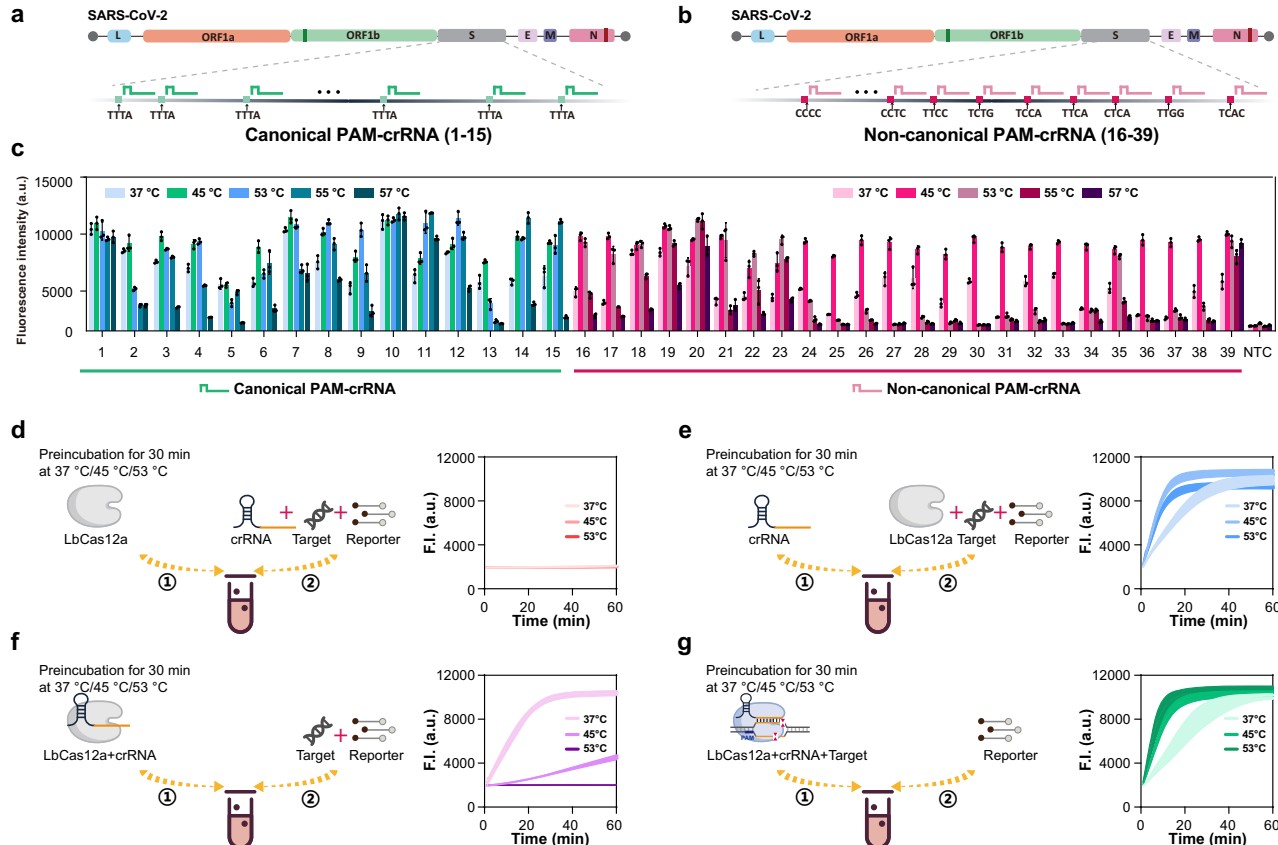

**Fig. 2 | High temperature boosted Cas12a trans-cleavage activity results from dsDNA target activation. a** Design of 15 crRNAs targeting the S gene of SARS-CoV-2 with canonical TTTA as PAM. Number these crRNAs from 1-15. **b** Design of 24 crRNAs targeting the S gene with non-canonical PAMs. Number these crRNAs from 16-39. **c** Evaluation of the *trans*-cleavage activity of different PAM-mediated activated LbCas12a at different temperatures. A target final concentration of 100 pM S-gene fragment was used, and the 39 crRNAs mentioned above were used to determine the collateral activity across the temperature range of 37 °C to 57 °C. Values are shown as endpoint fluorescence at 30 min. **d** Evaluation of the thermal stability of LbCas12a. LbCas12a was pre-incubated at 37 °C, 45 °C, or 53 °C for 30 min, after which the remaining components (crRNA, dsDNA target and reporter) were added for the 45 °C reaction and real-time fluorescence acquisition. **e** Evaluation of the thermal stability of crRNA. CrRNA was pre-incubated at 37 °C,

45 °C, or 53 °C for 30 min, after which the remaining components (LbCas12a, dsDNA target and reporter) were added for the 45 °C reaction and real-time fluorescence acquisition. **f** Evaluation of the thermal stability of LbCas12a-crRNA RNP. RNP was pre-incubated at 37 °C, 45 °C, or 53 °C for 30 min, after which the remaining components (dsDNA target and reporter) were added for the 45 °C reaction and real-time fluorescence acquisition. **g** Evaluation of the thermal stability of dsDNA target activated LbCas12a. LbCas12a-crRNA RNP and target DNA were pre-incubated at 37 °C, 45 °C, or 53 °C for 30 min, after which the reporter was added for the 45 °C reaction and real-time fluorescence acquisition. ① and ② represent the order of addition. Data are represented as mean ± standard error (*n* = 3 technical replicates). F.I. represents fluorescence intensity. a.u. represents arbitrary units. Source data are provided as a Source Data file.

and TTGG) (Fig. 3c, d). To clearly observe the two product bands after *cis*-cleavage, we treated the samples with proteinase K to prevent potential non-specific binding (Supplementary Fig. 14). In the electrophoresis experiments, we found that both canonical and non-canonical PAMs mediated *cis*-cleavage within the temperature range of 37 °C to 55 °C, but *cis*-cleavage efficiency was significantly higher with canonical PAMs. Increasing the reaction temperature to 45 °C did not improve *cis*-cleavage activity, indicating that the *cis*- and *trans*-cleavage activities of Cas12a are independent of each other[26]. To further support these observations, we quantified the relative *cis*-cleavage efficiencies from the electrophoretic bands using ImageJ and visualized the results as heatmaps (Fig. 3e–h). The quantitative analysis was consistent with the gel data, showing that canonical PAMs yielded higher cleavage efficiency than non-canonical PAMs across all tested temperatures and that increasing temperature did not enhance *cis*-cleavage activity. Together with the earlier *trans*-cleavage results, these observations indicate that elevating the reaction temperature predominantly enhances *trans*-cleavage activity. This effect likely reflects the intrinsic mechanistic differences between the two catalytic modes of Cas12a. During *cis*-cleavage, target recognition, PAM

engagement, and R-loop formation precisely orient the target strand into the RuvC catalytic pocket[23,27], making this process largely dictated by stable crRNA–DNA base pairing and structural positioning and therefore relatively insensitive to moderate temperature elevation. By contrast, *trans*-cleavage requires additional conformational rearrangements following R-loop stabilization to permit repeated access of freely diffusing single-stranded substrates to the active site. Increased temperature likely accelerates the diffusion and collision frequency of these trans substrates and facilitates their engagement with the catalytic pocket, thereby selectively boosting *trans*-cleavage efficiency without proportionally affecting *cis*-cleavage.

As the reaction temperature increased and the reaction time was extended, we observed a noticeable downward shift and smearing of both cleaved and uncleaved target DNA, eventually leading to complete degradation. Such abnormal migration patterns suggest that activated Cas12a may induce temperature-dependent degradation of double-stranded DNA. To confirm this, we used a 59 nt single-stranded DNA as the target to activate Cas12a, and then employed non-targeting 1000 bp double-stranded DNA as the substrate for temperature-dependent cleavage analysis (Supplementary Fig. 15). As shown, after

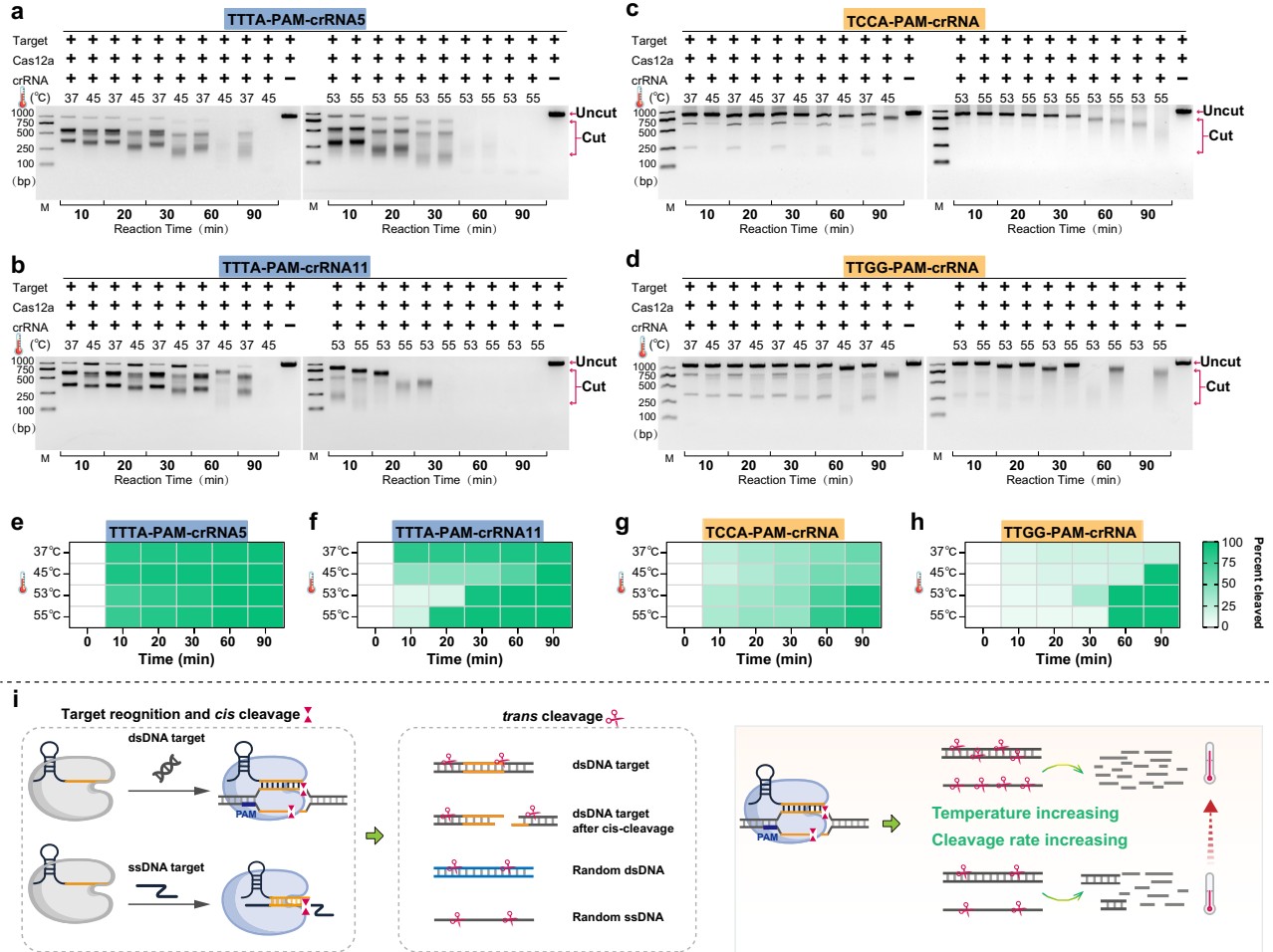

**Fig. 3 | Evaluation of the PAM-mediated dsDNA cleavage activities of Cas12a under different temperatures. a**, **b** In vitro dsDNA cleavage activities of LbCas12a mediated by canonical TTTA PAM. **c**, **d** In vitro dsDNA cleavage activities of LbCas12a mediated by non-canonical PAM (TCCA, TTGG). These four crRNAs and the dsDNA target used are from Fig. 2a, b. Double-stranded DNA cleavage activities were analyzed using 2% agarose gels at 10, 20, 30, 60 or 90 min across different temperatures (37–55 °C). The final concentration of dsDNA used was 120 nM. The experiment was independently repeated twice with similar results. bp represents base pairs. Source data are provided as a Source Data file. **e**–**h** Quantitative analysis of Cas12a *cis*-cleavage efficiencies under different temperatures. *Cis*-cleavage

efficiencies were calculated from the electrophoretic bands in **a**–**d** by densito-metric analysis using *Image J*. For each lane, the efficiency was determined as the difference between the reference lane (unreacted control) and the uncleaved target band, divided by the reference lane intensity. Heatmaps display the temperature-dependent *cis*-cleavage efficiencies of canonical PAMs (**e**, **f**) and non-canonical PAMs (**g**, **h**). **i** Schematic diagram of diverse substrates for *trans*-cleavage with activated Cas12a. When Cas12a is activated by target (ssDNA/dsDNA), it can *trans*-cleave dsDNA target, *cis*-cleaved DNA substrates, any double-stranded DNA, and any single-stranded DNA substrates. Additionally, this trans-cleaving activity increases with rising temperature. Source data are provided as a Source Data file.

60 min at 37 °C, only a slight shift and smearing of the target bands were observed. However, as the temperature increased, the downward shift and smearing of the target bands became more pronounced, confirming that activated Cas12a can efficiently *trans*-cleave and degrade blunt-ended double-stranded DNA at high temperatures.

Previous studies generally suggest that CRISPR-Cas12a exhibits little or no *trans*-cleavage activity against double-stranded DNA substrates[6], likely due to the difficulty of the RuvC catalytic pocket in accommodating large double-stranded DNA substrates. Recent research has proposed that Cas12a degrades double-stranded DNA directionally, preferentially *trans*-cleaving dsDNA with 3′ overhangs rather than 5′ overhangs[28]. This preference for overhang direction is thought to be related to the 3′–5′ exonuclease activity of Cas12a. However, these studies were conducted at 37 °C. In our study, when the reaction temperature was raised to 45 °C or higher, Cas12a's *trans*-cleavage activity against blunt-ended double-stranded DNA progressively increased. We speculate that this phenomenon results from enhanced DNA "breathing" at elevated temperatures, which transiently exposes single-stranded regions accessible to Cas12a's RuvC domain. Consequently, higher temperature increases the availability of

cleavable single-stranded sites within dsDNA, enabling Cas12a to degrade a broader range of *trans*-cleavage substrates. Therefore, we conclude that reaction temperature can modulate the kinetics of Cas12a's *cis*- and *trans*-cleavage activities. Specifically, 37 °C is optimal for *cis*-cleavage, while elevated temperatures enhance *trans*-cleavage activity. At higher temperatures, Cas12a can efficiently *trans*-cleave diverse substrates, including blunt-ended double-stranded DNA, sticky-ended double-stranded DNA, and single-stranded nucleic acids (Fig. 3i).

## PAM and temperature synergistically regulated CRISPR-Cas12a reaction improving nucleic acid detection

CRISPR diagnostics are often combined with isothermal nucleic acid amplification[8–10,18,29–31]. However, CRISPR cleavage activity can interfere with amplification reactions by degrading the template, presenting a challenge for developing highly sensitive one-pot CRISPR assays. The weak *cis*-cleavage activity of non-canonical PAMs has been exploited in the development of one-pot CRISPR detection (sPAMC)[18], but this comes at the cost of reduced detection efficiency due to the weak *trans*-cleavage activity under non-canonical PAM conditions. Our

study demonstrates that while *cis*-cleavage activity of non-canonical PAMs is weak at 37 °C, increasing the temperature to 45 °C or higher enhances their *trans*-cleavage activity to levels comparable to canonical PAMs. Based on these findings, we propose a poikilothermal one-pot CRISPR detection platform (POP-CRISPR) with improved nucleic acid detection. The key to this platform lies in the use of non-canonical PAMs and the implementation of a variable temperature program. At 37 °C, isothermal nucleic acid amplification predominates and is minimally affected by CRISPR cleavage. In contrast, at higher temperatures, such as 45 °C, CRISPR-based detection reaches optimal efficiency. To validate this hypothesis, we designed two crRNAs targeting the L1 gene of HPV-16 and HPV-18 based on the PAMs CCCC and CCTG, respectively, and performed sensitivity tests using the POP-CRISPR platform (Fig. 4a, b, and Supplementary Table 3). In parallel, one-pot isothermal methods based on the canonical PAM TTTA and the sPAMC method with non-canonical PAMs were also executed for comparison (Supplementary Fig. 16). The results showed that for both targets, the one-pot method with the canonical PAM TTTA exhibited low detection sensitivity. The sPAMC method significantly enhanced sensitivity, while POP-CRISPR demonstrated an order of magnitude lower detection limit than sPAMC (Fig. 4c, d). Furthermore, we validated the applicability of POP-CRISPR for RNA target detection using in vitro-transcribed SARS-CoV-2 N gene RNA (Supplementary Table 7), showing a detection limit of 1 aM with higher endpoint signal intensity and faster reaction kinetics than sPAMC (Supplementary Fig. 17).

We envision that further shortening the nucleic acid amplification time could accelerate detection speed. To test this, we set up a time gradient for the POP-CRISPR method at 37 °C with amplification times of 5, 10, and 30 min (Fig. 4e, f). We found that reducing the amplification time to 10 min resulted in higher detection efficiency compared to 30 min, while a 5-min amplification time yielded the lowest overall detection signal. Electrophoretic analysis further confirmed that amplicon accumulation nearly reached saturation after 10 min, with no significant increase observed at 30 min (Supplementary Fig. 18). Thus, for the POP-CRISPR method, a 10-min pre-amplification at 37 °C is sufficient to support high-sensitivity detection without causing signal delay, as prolonged amplification primarily postponed the initiation of highly efficient Cas12a *trans*-cleavage activity (Fig. 4g). Based on this optimized detection protocol, we conducted 20 independent measurements using HPV-16 DNA at final concentrations of 1 aM, 0.5 aM, and 0.2 aM (Supplementary Fig. 19). The results showed that at target concentrations as low as 1 aM (9 copies/reaction) and 0.5 aM (4.5 copies/reaction), POP-CRISPR achieved a 100% detection rate. At a target concentration of 0.2 aM (1.8 copies/reaction), the detection rate was 90%. A positive detection rate greater than 95% was defined as the detectable limit, establishing the limit of detection for POP-CRISPR at 4.5 copies/reaction.

Distinguishing highly similar sequences is critical in fields such as viral mutation detection, bacterial typing, and epigenetic analysis. CRISPR diagnostic technologies have been extensively engineered to enhance analytical specificity[7,32–35]. Strategies such as introducing artificial mutations[7] and the computational design of highly specific crRNAs[32] have been successful, but they still fall short of achieving single-base resolution detection. To our knowledge, the effect of reaction temperature on the recognition specificity of the CRISPR system has not been thoroughly studied. In this study, we generated a set of double-stranded DNA targets of identical length, containing both the canonical PAM-TTTA (Fig. 4h, i) and non-canonical PAM-GTTA (Fig. 4j, k). We also designed a group of crRNA variants incorporating consecutive two-base or single-base mutations at different positions in the paired spacer region. The results showed that, at three different temperatures (45, 53, and 55 °C), the detection efficiency of the Cas12a system was similar when the crRNA-target DNA was perfectly matched, confirming that the detection efficiency will not decrease at these temperatures. As the reaction temperature

increased, Cas12a's tolerance for mismatches between the target and crRNA gradually decreased. For canonical PAMs, Cas12a exhibited sensitivity to double-base mutations at 45 °C at certain positions. However, at 53 °C and above, consecutive double-base mutations within the 1–16 position range led to a complete loss of detection signal. We also introduced single-base mutations at positions sensitive to double-base mutations at 45 °C. Upon increasing the temperature to 55 °C, the sensitivity of the Cas12a system to single-base mismatches was significantly enhanced, with all 10 mismatch sites showing a complete loss of detection signal. For non-canonical PAMs, the temperature-mediated sensitivity to mismatches was even more pronounced. At all three tested temperatures, consecutive double-base mutations in positions 1–18 led to a complete loss of detection signal. When the temperature was increased to 53 °C or 55 °C, all single-base mismatch sites also displayed significant mismatch recognition sensitivity. Mechanistically, elevated temperatures decrease the thermodynamic stability of crRNA–target hybridization, and this destabilization is disproportionately amplified at mismatched positions, where hydrogen bonding is more easily disrupted. As a result, imperfect duplexes fail to maintain the stable R-loop required for efficient Cas12a activation. In addition, non-canonical PAMs intrinsically exhibit weaker affinity for Cas12a, thereby increasing the reliance on stringent crRNA–target complementarity for productive target engagement and cleavage. The combination of reduced hybrid stability and lower PAM-binding strength at higher temperatures substantially narrows the tolerance window for mismatches, effectively enabling single-nucleotide discrimination and further improving detection specificity (Supplementary Fig. 20). In summary, we conclude that non-canonical PAMs and elevated temperatures can synergistically improve the specificity of the Cas12a system.

## POP-CRISPR applied to clinical pathogens detection with improved sensitivity and specificity

In this study, we used HPV and MP as targets to demonstrate the clinical pathogen detection performance of POP-CRISPR. Persistent HPV infection, particularly with high-risk type HPV-16, is closely linked to the development of cervical cancer[36]. Rapid screening is critical for early intervention and controlling pathogen transmission. MP is a major causative agent of community-acquired pneumonia in children, making the rapid identification of MP infections essential for distinguishing them from other causes, such as bacterial or viral pneumonia[37]. This is crucial for timely targeted treatment and avoiding unnecessary antibiotic use. Here, we collected a total of 33 vaginal swab samples suspected of HPV-16 infection (Fig. 5a) and 65 nasopharyngeal swab samples suspected of MP infection (Fig. 5e). Nucleic acids were extracted from these samples, and the performance of POP-CRISPR was compared with two other one-pot reactions: one mediated by the canonical PAM TTTA and the other based on sPAMC (Supplementary Table 4). For HPV-16 detection, POP-CRISPR successfully identified 25 positive samples (*Ct* values: 11.9–35.7) and 7 negative samples (*Ct* > 37 considered negative), with results fully consistent with the gold-standard qPCR, achieving 100% concordance (Supplementary Table 8). The sPAMC method also detected all samples, but the signal-to-noise ratio was lower for samples with low concentrations. In contrast, the conventional TTTA-PAM one-pot method was only able to detect 10 positive samples (Fig. 5b–d). For MP detection, POP-CRISPR identified all 49 positive samples (*Ct* values: 23.5–36.2) and 16 negative samples, with results matching qPCR (Supplementary Table 9). However, sPAMC was only able to detect 43 positive samples, and the conventional PAM one-pot method failed to detect all positive samples (Fig. 5f–h). In summary, we demonstrated that POP-CRISPR outperforms the other two one-pot reactions in clinical pathogen detection, particularly for ultra-low concentration "gray zone" samples that are near the detection limit. POP-CRISPR consistently provides stable and more reliable results.

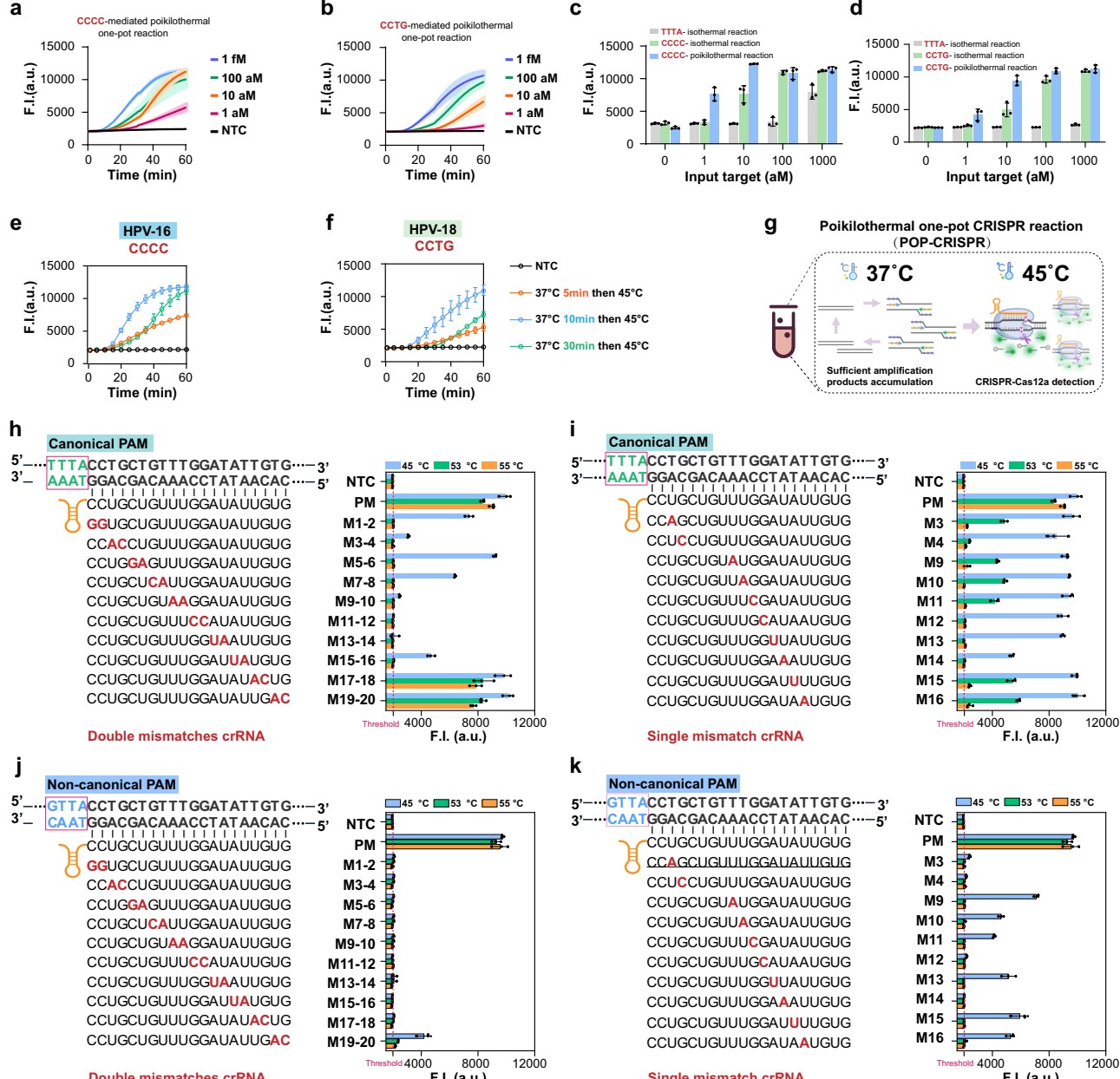

**Fig. 4 | Evaluation of the sensitivity and the specificity of the Cas12a system under PAM and temperature synergistically regulated CRISPR-Cas12a reaction. a** Non-canonical PAM (CCCC) mediated POP-CRISPR for the detection of HPV-16. **b** Non-canonical PAM (TTCC) mediated POP-CRISPR for the detection of HPV-18. The experimental procedure was to incubate at 37°C for 30 min and then to change the temperature to 45°C. **c** Comparison of HPV-16 detection sensitivity. Three one-pot methods (canonical PAM mediated isothermal reaction, sPAMC, and POP-CRISPR) were used to detect HPV-16 target at varying concentrations. **d** Comparison of HPV-18 detection sensitivity. Three one-pot methods (canonical PAM mediated isothermal reaction, sPAMC, and POP-CRISPR) were used to detect HPV-18 target at varying concentrations. Values are shown as endpoint fluorescence at 60 min. **e** POP-CRISPR amplification time optimization for HPV-16 detection. **f** POP-CRISPR amplification time optimization for HPV-18 detection. The final concentrations of the targets used in (**e**, **f**) were all 10 aM. **g** Schematic diagram of the principle of the POP-CRISPR. **h** Double bases mutation tolerance of Cas12a mediated by canonical PAM at varying temperatures. We designed crRNA variants with two consecutive mismatches (red) at different positions in the spacer. **i** Single base mutation tolerance of Cas12a mediated by canonical PAM at varying temperatures. The crRNA variants with single mismatch (red) were designed according to the spacer positions which were sensitive to double mismatches at 45°C. **j** Double bases mutation tolerance of Cas12a mediated by non-canonical PAM (GTTA) at varying temperatures. The crRNA variants used are those shown in (**h**). **k** Single base mutation tolerance of Cas12a mediated by non-canonical PAM (GTTA) at varying temperatures. The crRNA variants used are those shown in (**i**). Values are shown as endpoint fluorescence at 60 min. The threshold value (red dashed line) was set based on the mean fluorescence of no target control plus three times the standard deviation. Data are represented as mean ± standard error (*n* = 3 technical replicates). F.I. represents fluorescence intensity. a.u. represents arbitrary units. Source data are provided as a Source Data file.

Next, we aimed to evaluate whether POP-CRISPR could be applied for gene detection at single-base resolution. In recent years, the increasing rate of macrolide-resistant Mycoplasma pneumoniae infections (MRMPI) has become a global concern. MRMPI can prolong symptom duration and increase the risk of complications and sequelae[38,39]. Therefore, accurate and rapid detection of MRMPI is critical for effective clinical diagnosis and treatment. The most common mutation responsible for macrolide resistance is the A2063G mutation in the 23S rRNA gene, which accounts for over 90% of resistant strains. This mutation region lacks available classical PAMs,

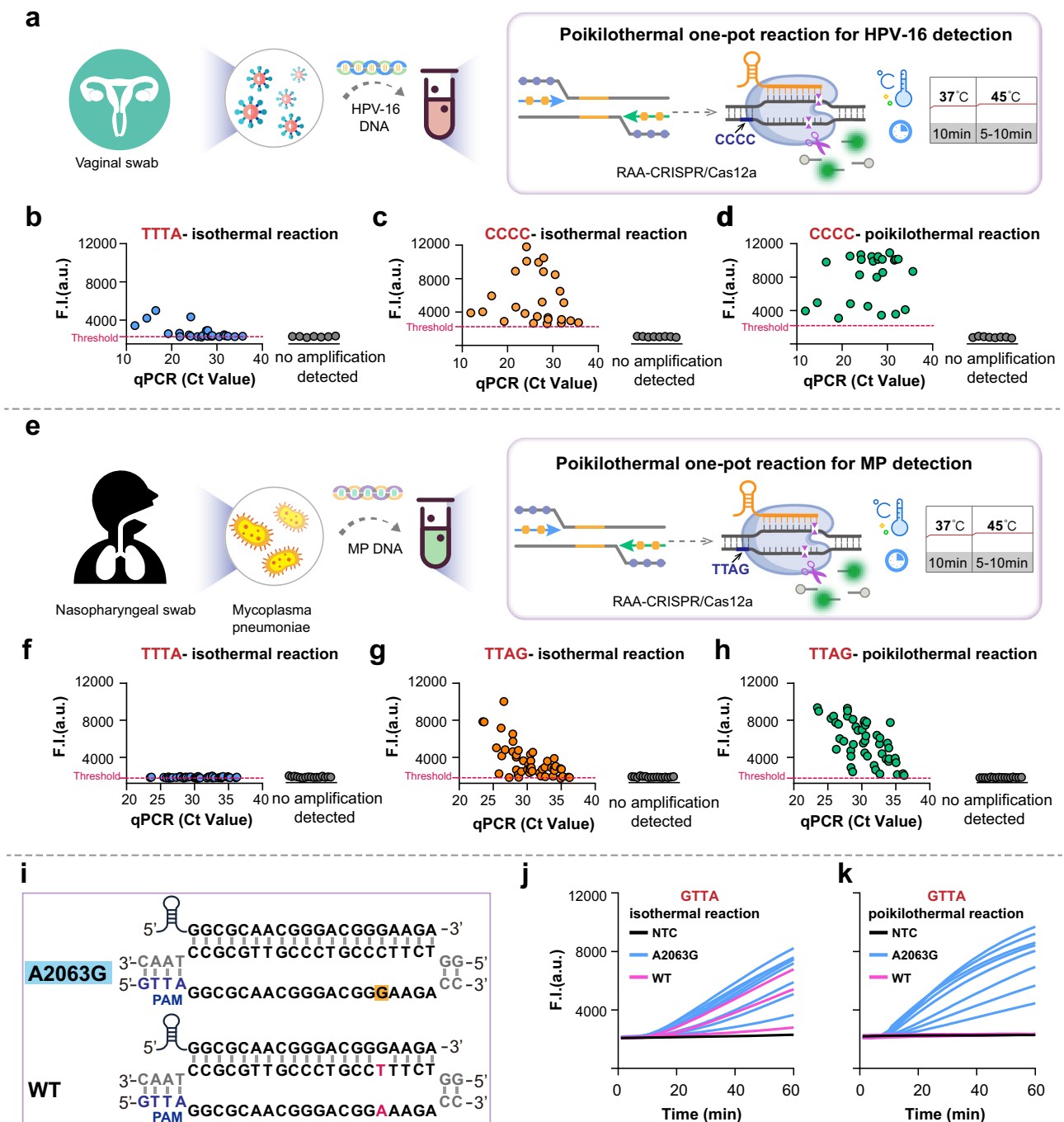

**Fig. 5 | Clinical application of POP-CRISPR for HPV-16 and MP diagnostics with improved sensitivity and specificity. a** Schematic diagram of clinical HPV-16 detection using POP-CRISPR. **b** Scatterplot of canonical PAM (TTTA)-mediated one-pot HPV-16 detection. **c** Scatterplot of sPAMC for HPV-16 detection. **d** Scatterplot of POP-CRISPR for HPV-16 detection. A total of 33 clinical HPV-16 nucleic acid samples were tested using these three assays. The coloured dots represent clinical samples that have been qPCR-identified as positive for HPV-16, while grey represent negative samples. **e** Schematic diagram of the clinical MP detection using POP-CRISPR. **f** Scatterplot of canonical PAM (TTTA)-mediated one-pot MP detection. **g** Scatterplot of sPAMC for MP detection. **h** Scatterplot of POP-CRISPR for MP detection. A total of 65 clinical MP nucleic acid samples were tested using these

three assays. The colored dots represent clinical samples that have been qPCR-identified as positive for MP, while grey represent negative samples. The threshold value (red dashed line) was set based on the mean fluorescence of negative samples plus three times the standard deviation. **i** The MP 23S RNA A2063G mutation and the wild-type sequence are presented. CrRNA was designed to be fully complementary to the A2063G mutant, while exhibits a base mismatch with the wild-type sequence at spacer position 16 (red). **j, k** sPAMc (37 °C) and POP-CRISPR (37 °C 10 min then 53 °C) were used to detect macrolide resistance mutation strains in 11 clinical MP nucleic acid samples. Blue lines represent A2063G mutation samples, red lines represent wild type samples. F.I. represents fluorescence intensity. a.u. represents arbitrary units. Source data are provided as a Source Data file.

making its detection challenging with conventional CRISPR-Cas12a diagnostic methods. We identified a non-canonical PAM (GTTA) near this mutation site and designed a crRNA that was perfectly complementary to the mutated strain, with only a single-base mismatch at position 16 in the wild-type strain (Fig. 5i). In the POP-CRISPR method,

we applied a 37–53 °C temperature program and tested 11 clinical samples confirmed to be infected with MP (Fig. 5k). The results showed that 8 samples tested positive using POP-CRISPR. Subsequent qPCR confirmed that these 8 samples carried the A2063G mutation (Supplementary Table 10), while the remaining 3 samples, which showed no

signal, were confirmed to be wild-type MP. Parallel testing with sPAMC detected all 11 samples but did not differentiate between resistant and wild-type strains as effectively (Fig. 5j). These results demonstrate that the high detection specificity of the POP-CRISPR method enables successful single-base resolution detection in clinical applications.

## Development of rapid samples processing procedure and mini-device for on-site testing application

Next, we are developing POP-CRISPR for on-site pathogen detection (Fig. 6a). Most nucleic acid detection methods for pathogen diagnosis require multi-step and time-consuming nucleic acid extraction (typically taking over 20 min), which is impractical for point-of-care testing (POCT)[40]. Chelex-100 resin, a heat-stable styrene-divinylbenzene copolymer containing iminodiacetic acid ions, can chelate divalent metal ions (such as calcium and magnesium), effectively inhibiting nuclease activity[41,42]. Chelex-100 has been widely used for rapid nucleic acid processing in biological samples[18,29,43]. To develop a rapid nucleic acid release protocol suitable for POP-CRISPR, we optimized several key conditions, including the lysis buffer formulation, the ratio of sample to buffer, heating temperature, and lysis time. We validated the nucleic acid release efficiency from nasopharyngeal swab clinical samples under different experimental conditions using qPCR (Supplementary Fig. 21). Ultimately, we identified the optimal lysis buffer and conditions: a 20% Chelex-100 TE buffer (pH 8.0), 50 mM DTT, a 1:1 sample-to-lysis buffer ratio, and a 2-min heat treatment at 80 °C. These conditions provided high-quality nucleic acids for downstream detection.

We collected 27 clinical nasopharyngeal swab samples suspected of MP infection and performed rapid nucleic acid release using the aforementioned Chelex-100-based lysis buffer. For comparison, we also used a commercially available column-based extraction kit to perform standardized nucleic acid extraction on the same samples as a control. The commercial kit required over 20 min of operation, involving multiple centrifugations and liquid transfers, while the Chelex-100-mediated heat lysis method only required a single-tube, one-step operation that took just 2 min. Next, we used POP-CRISPR to validate the effectiveness of both nucleic acid extraction methods. As shown in Fig. 6b, both extraction methods successfully detected all 13 positive samples, producing similar fluorescence signal intensities, with no false positives detected in the negative samples. These results were fully consistent with qPCR outcomes. This demonstrates that the rapid nucleic acid release method significantly shortens sample processing time while achieving extraction efficiency comparable to that of commercial kits. Additionally, to assess the stability of nucleic acids extracted using this rapid lysis method, we stored the processed samples at −20 °C for 1 month and repeated the experiments. The results were almost consistent with those obtained from immediate processing (Supplementary Fig. 22).

Finally, to overcome the limitations of traditional detection methods and enable portable, rapid, on-site diagnostic testing, we developed a compact, portable detection device capable of flexible temperature control and efficient fluorescence signal collection (Fig. 6a). This device integrates several advanced components, including aluminum alloy heating blocks, custom PCB heat shields, filter retention plates, magnets, and various electronic elements. The design schematic is illustrated in Supplemental Fig. 23. The mini-device can be connected to a smartphone application via Bluetooth, allowing real-time display of data curves on the home screen and seamless interaction with the device. Users can customize heating profiles by adjusting parameters such as temperature, duration, and step additions, enabling automatic temperature adjustments tailored to newly developed assays. After the reaction, the mini-device autonomously analyzes the results and provides a positive or negative outcome based on a preset threshold. This makes the device ideal for resource-limited field settings and on-site detection across various scenarios. Using this portable instrument (Supplementary Fig. 24), we tested 27 nasopharyngeal swab samples, achieving 100% detection sensitivity and specificity (Supplementary Table 11). The testing time for each sample can be controlled within 20 min. Notably, the recorded real-time fluorescence curves demonstrated that the fluorescence detection efficiency for both negative and positive samples was comparable to that of quantitative PCR fluorescence detection instruments (Supplementary Fig. 25).

## Discussion

The widely used CRISPR-Cas12a system relies on the short PAM motif "TTTV" as a prerequisite for recognizing and cleaving target DNA[20]. This inherent PAM requirement restricts the selectivity of detectable targets, especially in applications such as antibiotic resistance or tumor mutation detection, where mutations may not always be adjacent to classical TTTV PAM sites. Although several non-canonical PAM sites have been identified within the CRISPR-Cas12a system[13,17,44,45], their occurrence remains limited, and their cleavage activity is generally much lower than that observed with canonical PAMs. In this study, we comprehensively evaluated the effects of all 256 possible PAM sites (NNNN, where N = A, T, G, C) on Cas12a trans-cleavage activity. We demonstrated that, under high-temperature reaction conditions, up to 82 PAM sites exhibited trans-cleavage efficiencies comparable to those of canonical PAMs, suggesting that a significant number of non-canonical PAM sites could be utilized for target design. This expands the available target selection options, offering more flexibility in detection design without being constrained by TTTV PAMs. Interestingly, high reaction temperatures enhance trans-cleavage efficiency but do not mediate cis-cleavage efficiency enhancement; in fact, they impair it. This indicates that the two cleavage mechanisms have independent temperature dependencies. Further investigation revealed that the LbCas12a system can tolerate temperatures as high as nearly 60 °C without significant loss of trans-cleavage activity. To our knowledge, no prior reports have indicated that wild-type, unengineering Cas12a can maintain trans-cleavage activity at such high temperatures. This high-temperature adaptability may be valuable for developing new CRISPR-based biotechnologies, such as those used in nucleic acid detection, where higher-temperature amplification methods can be integrated.

Previous studies have exploited the weak cis-cleavage activity of non-canonical PAMs to develop the one-pot CRISPR detection system, sPAMC[18]. This system reduces cleavage of the nucleic acid amplification template, thereby balancing both nucleic acid amplification and CRISPR detection efficiency. In this study, we present the POP-CRISPR diagnostic platform, which leverages low cis-cleavage activity at 37 °C and efficient trans-cleavage activity at elevated temperatures for non-canonical PAM sites. Compared to sPAMC, POP-CRISPR requires only a simple poikilothermal detection program, resulting in approximately an order-of-magnitude improvement in detection sensitivity. This enhancement is crucial for improving the signal-to-noise ratio and detection sensitivity in low-concentration biological samples. Using a large set of clinical samples from HPV-16 and MP, we demonstrated the value of POP-CRISPR in boosting detection efficiency.

At 37 °C, both cis- and trans-cleavage efficiencies of LbCas12a under non-canonical PAM conditions are low, prompting us to explore how optimizing the temperature program could enhance both CRISPR detection efficiency and speed. Interestingly, we found that a 10-min nucleic acid amplification step at 37 °C, followed by CRISPR-Cas12a detection at 45 °C, resulted in the highest detection sensitivity, compared to longer amplification times, such as 30 min. This finding challenges the common belief that extended amplification times lead to higher detection sensitivity. We hypothesize that isothermal amplification progresses rapidly in the early stages, and once a sufficient amount of product is accumulated, the efficient trans-cleavage signal amplification by the Cas12a system becomes the key factor.

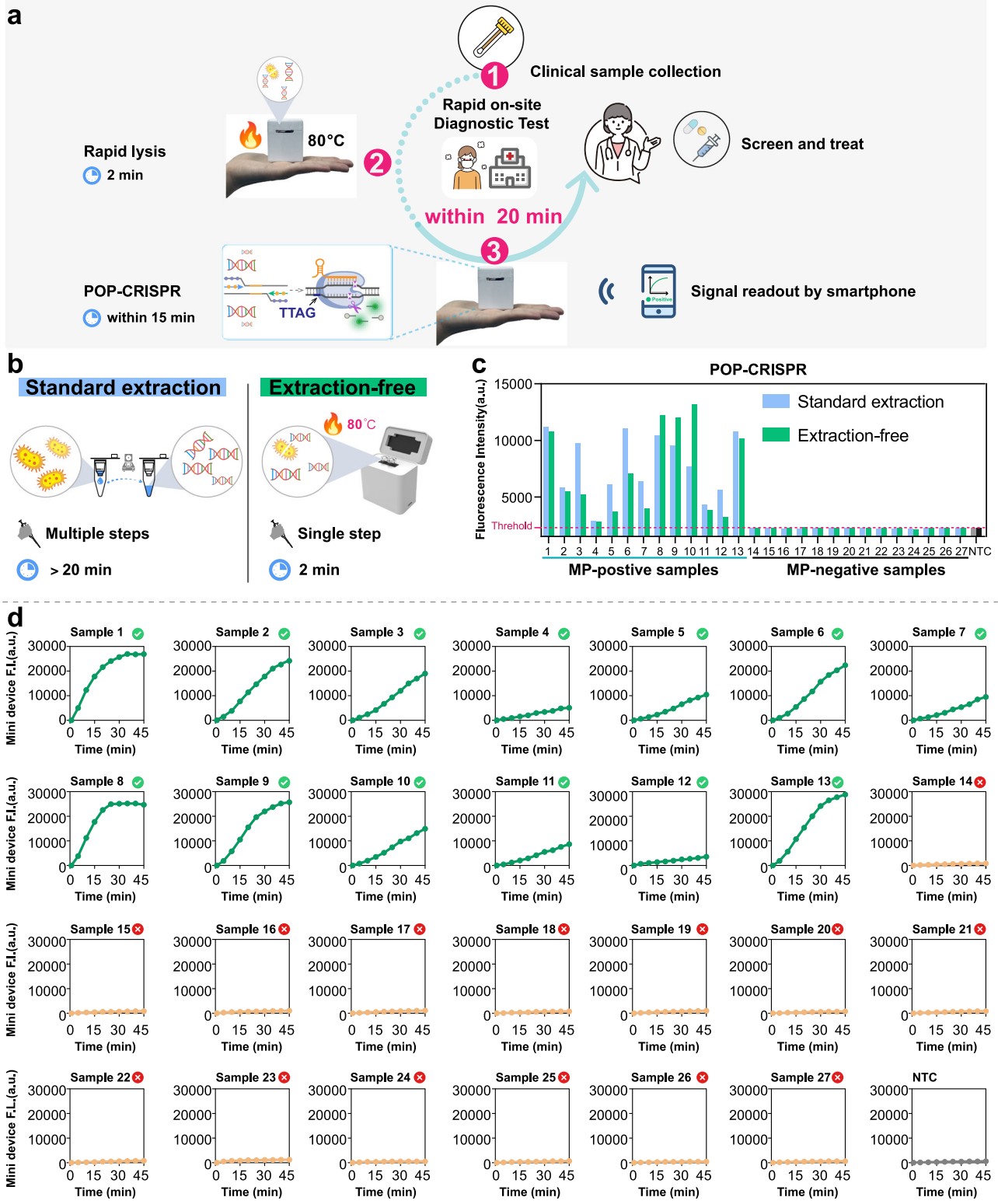

Therefore, initiating CRISPR detection earlier may be more beneficial for improving detection efficiency. In conclusion, these findings provide a strategy to enhance both the sensitivity and speed of CRISPR-based diagnostic technologies.

In addition to the improved detection efficiency, we found that another significant advantage of POP-CRISPR is the marked enhancement in detection specificity compared to existing CRISPR diagnostic technologies and qPCR (Supplementary Table 12). Previous efforts to improve the specificity of CRISPR diagnostics have focused on the design and engineering of crRNAs[34,35,46], yet single-base resolution detection remains unsatisfactory. Temperature, a critical physical parameter, has been used to develop highly specific nucleic acid recognition methods, such as melt curve analysis in PCR[47]. In this study, we explored the impact of temperature on CRISPR-Cas12a specificity and found that increasing the reaction temperature above 45 °C significantly enhanced the system's ability to distinguish highly similar sequences, particularly at non-canonical PAM sites. We observed that even a single-base mismatch at many positions resulted

**Fig. 6 | On-site detection of MP using POP-CRISPR. a** Workflow of POP-CRISPR for on-site testing for MP using mini-device. Green arrows and red numbers (①–③) indicate the three main steps and their sequential order in the detection workflow. First, nasopharyngeal swab clinical samples are collected and mixed with Chelex-100-containing lysis buffer, followed by rapid nucleic acid release on the mini-device at 80 °C for 2 min. The obtained nucleic acids are then added to the POP-CRISPR reaction system, which is performed on the mini-device using a poikilothermal one-pot protocol (37 °C for 10 min followed by 45 °C). The detection results can be interpreted via wireless connection to a smartphone application, serving as a reference for subsequent diagnosis and prognosis. Selected icons in this figure were created by soco-st and are available on SVGRepo.com (https://www.svgrepo.com/svg/492696/online-medical-treatment-female; https://www.svgrepo.com/svg/493281/fever-female). These icons are licensed under the Creative Commons Attribution 4.0 International License (CC BY 4.0) (https://creativecommons.org/licenses/by/4.0/), in accordance with the SVGRepo licensing terms (https://www.svgrepo.com/page/licensing/#CC%20Attribution). **b** Scheme for standardized nucleic acid extraction and rapid lysis for nucleic acid release. **c** Evaluation the extraction efficiency of two nucleic acid extraction methods using POP-CRISPR. 27 clinical nasopharyngeal swab samples suspected of MP infection were tested. Values are shown as endpoint fluorescence at 60 min. The threshold value (red dashed line) was set based on the mean fluorescence of no target control plus three times the standard deviation. **d** Real-time fluorescence profiles obtained from POP-CRISPR experiments performed on mini-device. The experimental procedure was as follows: 10 min at 37 °C followed by 45 °C, figure shows the real-time fluorescence signal of 45 °C. F.I. represents fluorescence intensity. a.u. represents arbitrary units. Source data are provided as a Source Data file.

in the complete loss of detectable signal. This is a crucial finding, as single-base resolution detection is essential in fields such as tumor gene mutation analysis, viral variant detection, bacterial genotyping, antibiotic resistance gene identification, and epigenetic studies. To evaluate the clinical performance of this enhanced specificity, we applied the POP-CRISPR platform to analyze 11 clinical MP samples. The method reliably detected the A2063G mutation, which confers macrolide antibiotic resistance, in 8 samples, while no detectable signal was observed in the wild-type samples. In contrast, when we tested the same samples using sPAMC, distinguishing between wild-type and mutant strains proved difficult. These results highlight the superior specificity of the POP-CRISPR method, making it highly suitable for applications requiring single-base resolution detection in clinical diagnostics.

In summary, by increasing the reaction temperature, we identified a broad range of PAM sites that can be applied to nucleic acid detection. Through investigating the synergistic regulation of *cis*- and *trans*-cleavage activities in the LbCas12a system by temperature and PAM, we developed the POP-CRISPR diagnostic platform. POP-CRISPR has been successfully applied to detect a wide variety of clinical pathogens, demonstrating significant advantages in detection sensitivity, speed, specificity, and a broader range of target selection. Furthermore, by simplifying the sample preparation process and developing a portable device, we established a complete workflow—from clinical pathogen sample input to detection signal acquisition—within 20 min, confirming the potential of POP-CRISPR for POCT applications.

## Methods

### Ethical statement

Clinical samples for this research were collected from the first hospital of Guangzhou Medical University. All participants provided informed consent prior to sample collection, demonstrating a full understanding of the study's purpose, procedures, and potential risks. Participants' personal information will be maintained in strict confidentiality, and all data will be processed anonymously to ensure the protection of their privacy. Samples were obtained by the research team in a deidentified manner, and no demographic information (including sex, age, gender, or other identifiers) was collected or made available to ensure strict participant privacy protection. The research protocol was performed following the guidelines approved by Ethics Committee of (approval ID: 2021-K-40).

### Materials

All the DNA sequences were synthesized by Sangon (Shangai, China), all RNA sequences were synthesized by Hippo Biotechnology (Huzhou, China). All plasmids were synthesized by Tsingke Biotech (Beijing, China). Detailed sequence information is provided in Supplementary Table 5. The RAA kit was purchased from Qitian Gene Biotechnology (Wuxi, China). The Cas12a and reaction buffer were obtained from Biolifesci (Guangzhou, China). Agarose gel recovery kit was purchased from Omega Bio-Tek. The HPV-16 qPCR kit and the *Mycoplasma pneumoniae* qPCR kit were obtained from Rongjin Technology (Shenzhen, China). Chelex 100 resin was purchased from BIO-RAD.

### Preparation of target library included all PAM sites

Using a plasmid containing the ASFV P72 gene fragment as a template, a PAM site with diverse nucleotide combinations (NNNN, where N = A/T/C/G) was designed for insertion at the middle position of the forward primer. PCR amplification was performed using the same reverse primer to obtain 256 double-stranded DNA targets, each 294 bp in length, containing different PAM sites. The 150 μL PCR amplification system comprised 1× Taq mixture, 300 nM forward primer, 300 nM reverse primer, and 10 ng/μL plasmid target. The PCR procedure consisted of the following steps: 94 °C for 1 min, 98 °C for 10 s, 55 °C for 5 s, and 72 °C for 30 s, repeated for a total of 35 cycles, followed by a final extension at 72 °C for 10 min. The PCR products were purified and recovered using an agarose gel recovery kit. The final products were accurately quantified with an Invitrogen Qubit fluorometer (Thermo Fisher scientific), diluted to a uniform concentration, aliquoted, and stored at −80 °C for subsequent use.

### In vitro CRISPR-Cas12a trans-cleavage experiments

The 20 μL reaction system comprised 1× Cas12a buffer, 500 nM polyC$_6$ (FAM−6C−BHQ1) fluorescent probe, 10 nM crRNA, 10 nM Cas12a, and varying final concentrations of DNA targets. The Bio-Rad CFX96 Touch Real-Time PCR System was utilized for thermostated reactions and the real-time acquisition of fluorescence signals, with data points collected at 1-min intervals.

### In vitro CRISPR-Cas12a cis-cleavage experiments

The 20 μL reaction system consisted of 1×Cas12a buffer, 300 nM crRNA, 300 nM LbCas12a, and 100 nM double-stranded DNA target. The reaction was incubated at 37, 45, 53, and 55 °C for 10, 20, 30, 60, and 90 min, respectively, and was terminated by the addition of 2 μL proteinase K. The results of cis-cleavage were analyzed using 2% agarose gel electrophoresis.

### Evaluation of the thermal stability of LbCas12a reaction components

Thermal stability assays were carried out in a 20 μL reaction system containing 1× Cas12a buffer (5 mM Tris-HCl, pH 9.0; 5 mM NaCl; 15 mM MgCl$_2$; 0.01% CA-630), 10 nM LbCas12a, 10 nM crRNA, 500 nM polyC$_6$ reporter (FAM−6C−BHQ1), and 100 pM dsDNA targets. For each test, the component under evaluation (LbCas12a, crRNA, pre-assembled LbCas12a−crRNA RNP, or RNP pre-activated with target DNA) was pre-incubated in 1× buffer supplemented with nuclease-free water at 37 °C, 45 °C, or 53 °C for 30 min. After pre-incubation, the remaining components were added to complete the 20 μL reaction, which was then incubated at 45 °C for real-time fluorescence acquisition using the Bio-Rad CFX96 Touch Real-Time PCR System, with data points collected at 1-min intervals.

### One-pot RAA-Cas12a isothermal reaction

The 50 μL reaction system consisted of 1× Cas12a buffer, 10 nM crRNA, 10 nM LbCas12a, 1 μM polyC$_6$ (FAM−6C−BHQ1) fluorescent probe, 250 nM of two pairs of primers, 25 μL RAA-Primer-Free Rehydration Buffer, 14 mM magnesium acetate, and 7.5 μL DNA target. The mixture was incubated at 37 °C for 60 min. The reaction was monitored using the Bio-Rad CFX96 Touch Real-Time PCR System.

### Poikilothermal one-pot CRISPR reaction (POP-CRISPR) for DNA target

The 50 μL reaction system consisted of 1× Cas12a buffer, 10 nM crRNA, 10 nM LbCas12a, 1 μM polyC$_6$ (FAM−6C−BHQ1) fluorescent probe, 250 nM of two pairs of primers, 25 μL RAA-Primer-Free Rehydration Buffer, 14 mM magnesium acetate, and 7.5 μL DNA target. The mixture was incubated at 37 °C for 10 min, followed by incubation at 45 °C for 50 min. The reaction was monitored using the Bio-Rad CFX96 Touch Real-Time PCR System.

### Poikilothermal one-pot CRISPR reaction (POP-CRISPR) for RNA target

The 50 μL reaction system consisted of 1× Cas12a buffer, 10 nM crRNA, 10 nM LbCas12a, 1 μM polyC$_6$ (FAM−6C−BHQ1) fluorescent probe, 400 nM of forward primer, 200 nM of reverse primer, 400 nM of RT-primer, 25 μL RAA-Primer-Free Rehydration Buffer, 14 mM magnesium acetate, 0.9 μL RNase H (50 U/μL, New England Biolabs, NEB), 0.45 μL SuperScript IV reverse transcriptase (Thermo Fisher) and 7.5 μL RNA target. The mixture was incubated at 37 °C for 10 min, followed by incubation at 45 °C for 50 min. The reaction was monitored using the Bio-Rad CFX96 Touch Real-Time PCR System.

### Analysis of clinical HPV-16 sample by qPCR

The 40 μL reaction system contained 23 μL PCR reaction premix, 2 μL PCR enzyme mixture, and 10 μL clinical sample. The following PCR program was used: 50 °C ×3 min (1 cycle), 95 °C ×2 min (1 cycle), 45 cycles of 95 °C ×5 s, 60 °C ×35 s and 72 °C ×45 s. The criteria in this qPCR kit was claimed as: Ct value ≤ 37 is positive, Ct value > 37 is negative.

### Analysis of clinical *Mycoplasma pneumoniae* sample by qPCR

The 30 μL reaction system contained 18.5 μL PCR reaction premix, 1.5 μL PCR enzyme mixture, and 15 μL clinical sample. The following PCR program was used: 50 °C ×2 min (1 cycle), 95 °C ×3 min (1 cycle), 45 cycles of 95 °C ×10 s and 60 °C ×30 s. The criteria in this qPCR kit was claimed as: Ct value ≤ 37 is positive, Ct value >37 is negative.

### Analysis of *Mycoplasma pneumoniae* drug resistance mutation sample by qPCR

The 25 μL reaction system contained 8 μL PCR reaction premix, 0.5 μL PCR enzyme mixture, and 5 μL clinical sample. The following PCR program was used: 50 °C ×2 min (1 cycle), 95 °C ×2 min (1 cycle), 45 cycles of 91 °C ×15 s and 64 °C ×1 min. The criteria in this qPCR kit was claimed as: FAM Ct<35.33, VIC < 35.01, A2063G mutation; FAM Ct≥35.33 or UNDET, VIC < 35.01, WT.

### Rapid lysis protocol

The lysis buffer was prepared as follows: 20% Chelex 100 resin (w/v), TE buffer (pH 8.0), and 50 mM DTT. Nasopharyngeal swab samples were mixed 1:1 with lysis buffer and heated at 80 °C for 2 min.

### Nucleic acid extraction of *Mycoplasma pneumoniae* from Nasopharyngeal swab samples

As a control for the rapid lysis protocol, we used a commercially available viral DNA extraction kit (Omega). This kit was employed for nucleic acid extraction from inactivated clinical samples which were collected from nasopharyngeal swabs of suspected *Mycoplasma pneumoniae* infections. The extraction was performed in the P2 laboratory at the First Hospital of Guangzhou Medical University, following the manufacturer's instructions.

### Overall design of the mini-device

The mini-device device is composed of four core modules, all efficiently integrated and managed by a main control PCB board with an ultra-low power consumption design. These modules include: (1) an optical module for highly sensitive fluorescence sensing and signal amplification; (2) a thermal module ensuring precise temperature regulation for reliable molecular detection; (3) a power management unit optimized for energy efficiency during operation; (4) a connectivity module enabling Internet of Things functionality for enhanced accessibility. Initial prototyping was achieved using nylon material via 3D printing with the Flashforge Guider II. Subsequent refinement was carried out through CNC machining to ensure precise component alignment and ease of installation. The aluminum alloy heating block was designed in SolidWorks and fabricated using CNC technology. Injection molding was employed to produce high-pass filters with precise dimensions of 2 mm in thickness, length, and width. A dedicated smartphone application was developed in Java using Android Studio, compatible with Android 8 and later operating systems.

### Statistics and reproducibility

No statistical method was used to predetermine sample size. No data were excluded from the analyses. The experiments were not randomized. The investigators were not blinded to allocation during experiments and outcome assessment.

### Reporting summary

Further information on research design is available in the Nature Portfolio Reporting Summary linked to this article.

## Data availability

All data supporting the conclusions of this study are included in the main text and Supplementary Information. Source data are provided with this paper.

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

## Acknowledgements

This work was supported by National Key R&D Program of China (2023YFC2307400), grants from the National Natural Science Foundation of China (32150019, 82502830), the Basic and Applied Basic Research Foundation of Guangdong Province (2021A1515220164, 2023A1515220160), China Postdoctoral Science Foundation (2023M741238), Guangdong Basic and Applied Basic Research Foundation Youth Fund project (2023A1515111076) and 2023 South China Normal University Young Teachers and Cultivation Fund project support (23KJ13).

## Author contributions

T.T. and X.M.Z. conceived and designed the study. T.Z., W.T.Z., and M.L. conducted the experiments. T.T., Z.T., and Z.Q.Q. contributed to the data analysis and interpretation. T.T. and X.M.Z. wrote the manuscript. B.Q.S., M.C., and B.X. collected and provided the clinical samples. C.Y.X. and Q.W.W. assisted with sample processing and nucleic acid extraction. Y.T.S., X.Y.G., M.E.H., and X.H.R. contributed to the construction of comprehensive PAM target library. J.Z.S. and F.B.P. designed and

constructed the mini-device. X.M.Z., B.Q.S., and M.C. supervised the manuscript. All authors reviewed and approved the final manuscript.

## Competing interests

J.Z.S. is a scientific advisor for Hangzhou EzDx Technology. The other authors declare no competing interests.
