## [Transparent Peer Review file · Nature Communications]

Identification of Thermotolerant Non-canonical PAMs for Robust One-Pot CRISPR-Cas12a Detection

Corresponding Author: Professor Xiaoming Zhou

Version 0:

Reviewer comments:

Reviewer #1

(Remarks to the Author)

Tian et al. systematically screened CRISPR-Cas12a PAM sites under elevated temperatures and identified over 80 non-canonical PAMs that exhibit trans-cleavage activities comparable to those mediated by canonical PAMs. In exploring the interplay between temperature and PAM identity in regulating Cas12a enzymatic activity and target specificity, the authors reported several notable findings that expand our understanding of the CRISPR-Cas12a system. For example, although trans-cleavage activity driven by non-canonical PAMs is markedly enhanced at elevated temperatures, the corresponding cis-cleavage remains relatively weak. This observation provides new evidence for the mechanistic decoupling of cis- and trans-cleavage activities in Cas12a. The authors further demonstrate that elevated temperature and non-canonical PAMs can act synergistically to improve target discrimination, achieving single-nucleotide resolution in SNP detection.

Building on these mechanistic insights, the authors developed a poikilothermal one-pot CRISPR platform based on thermotolerant non-canonical PAMs. This platform outperforms previous systems such as sPAMC in terms of sensitivity, specificity, and target flexibility. The applicability of POP-CRISPR for on-site detection was further demonstrated by optimizing sample processing protocols and integrating the assay with portable instrumentation. As such, this study integrates fundamental characterization of Cas12a biochemical behavior with the development of a practical diagnostic tool. Moreover, the synergistic enhancement of Cas12a activity and sequence specificity by temperature and PAM identity may be generalizable to other effectors in the Cas12 family, including Cas12b and Cas12f. Overall, the findings represent a meaningful contribution to both CRISPR biology and its diagnostic applications.

This is an interesting and valuable work, I strongly support publication of this manuscript in Nature Communications after addressing the following issues.

Major questions:

1. One of the key findings of this work is that the trans-cleavage activity of Cas12a is significantly enhanced at elevated temperatures, leading to the identification of over 80 previously unreported non-canonical PAMs with efficiencies comparable to the canonical ones. However, I noticed that the enhancement in trans-cleavage efficiency with increasing temperature appears to be more pronounced for crRNAs targeting non-canonical PAMs than for those targeting canonical PAMs (e.g., Figure 1). What might be the underlying reason for this difference? I believe a discussion of this point would not only strengthen the rigor of the conclusions but also enhance the reader's understanding of the work.
2. In the experiments evaluating the temperature tolerance of different crRNAs (e.g., Figure 2c), the authors observed that for both canonical and non-canonical PAMs, the optimal reaction temperature was consistently around 45 °C, with some crRNAs retaining considerable trans-cleavage activity even at temperatures as high as 57 °C. However, these experiments were all performed using double-stranded DNA (dsDNA) targets. Interestingly, when the targets were switched to corresponding single-stranded DNA (ssDNA), the optimal reaction temperature dropped to 37 °C, and increasing the temperature to 45 °C or higher led to a decrease in trans-cleavage activity (Supplementary Figure 7). What could be the mechanistic explanation for this discrepancy in temperature dependence between dsDNA and ssDNA targets? A discussion of this observation would enhance the understanding of the temperature-dependent behavior of Cas12a.
3. The authors mention that monomeric Cas12a protein is thermosensitive and can be nearly inactivated after 30 minutes of incubation at 37 °C. However, once activated by the target, the Cas12a RNP complex retains robust trans-cleavage activity even at 53 °C. I have two questions regarding this observation: (1) Was the crRNA used in this experiment itself one of the heat-tolerant crRNAs identified earlier? If a less heat-tolerant crRNA had been used, would the results have been different? (2) What might be the underlying reason for the improved thermal stability of target-activated RNPs compared to the

monomeric Cas12a protein? Could this be due to conformational stabilization upon target binding? I suggest that the authors consider including more direct experimental evidence—such as differential scanning fluorimetry (DSF)—to further explore this aspect.

4. In Lines 226–238, the authors report an interesting observation: experimental results demonstrate that increasing the reaction temperature enhances the trans-cleavage activity of Cas12a to the extent that it can even degrade blunt-ended double-stranded DNA—a phenomenon not previously reported. However, this section only describes the experimental outcome without offering any discussion or analysis of the underlying mechanism. I recommend that the authors include an appropriate and objective discussion on the possible reasons why Cas12a is capable of efficiently degrading a broader range of trans-cleavage substrates at elevated temperatures.

Minor questions:

1. How were the scatter plot data in Figures 1c and 1d generated? The authors should clarify how these data were calculated or derived. A brief description in the figure legend or the Methods section would enhance reproducibility.
2. In Figure 1, the PAM screening was performed using a ASFV P72 gene fragment, whereas Figures 4a–g focus on the detection of HPV16 and HPV18. Were the non-canonical PAMs used in these assays (such as CCCC and CCTG) also experimentally validated with the corresponding HPV targets? More generally, are the over 80 identified suboptimal non-canonical PAMs broadly applicable across different target sequences, or is their effectiveness sequence-dependent?
3. Although the scatter plots in Figure 1 show that the number of non-canonical PAMs with increased reaction rates grows substantially with rising temperature, this presentation only provides an overall view of the data and does not link the results to specific PAM sites. To facilitate future reference and selection of suitable non-canonical PAMs by readers, I suggest that the authors consider including the fluorescence reaction rate data corresponding to individual PAM sites in the supplementary materials.
4. When assessing the effect of temperature on Cas12a enzymatic activity (e.g., in Figure 2c), the authors selected 37, 45, 53, 55, and 57°C. What was the rationale for selecting these specific temperature points? The author should explain the reasoning for selecting these specific temperature values. Was this based on enzyme kinetics, thermostability, or empirical observations?
5. I noticed that in the experiments assessing Cas12a trans-cleavage activity at different temperatures, the concentration of the Cas12a–crRNA RNP complex was consistently set at 10 nM. I am curious whether increasing the RNP concentration might further improve the thermal tolerance of the system. I suggest the authors explore this possibility, as it could provide useful insights into optimizing reaction conditions under elevated temperatures.
6. In lines 214–216, the authors state that : “...the cis- and trans-cleavage activities of Cas12a are independent of each other.” A recent preprint (bioRxiv 2025.03.23.644851 ; doi: <https://doi.org/10.1101/2025.03.23.644851>) reports similar findings. The authors are encouraged to cite this reference to support their claim.
7. In Figures 4h to 4k, would it be possible to add individual data points to the bar plots? Doing so would allow readers to better assess the variability across replicates. The same issue is also present in Supplementary Figures 6 to 8, and I recommend that the authors consider making this adjustment consistently across all relevant figures.
8. Figures 4h and 4j include double-base substitutions across all 20 spacer positions, but Figures 4i and 4k only show selected single-nucleotide variants. How were the positions for single-nucleotide substitutions chosen, and could additional explanation be given?
9. The authors employed a Chelex-100-based thermal lysis protocol for nasopharyngeal swabs (Figures 6b–c). Could this method be extended to other sample types such as cervical swabs or blood-derived fluids?
10. There are a few typographical and formatting issues that merit attention. For example, the term “trans” in line 119 should be properly formatted, and the label in Figure 5a should read “RAA-CRISPR/Cas12a” instead of “RRA-CRISPR/Cas12a”.
11. In Supplementary Table S5, the lack of dividing lines makes it difficult to distinguish between the two plasmid sequences. I recommend that the authors carefully check and reformat the table to clearly separate the sequences for clarity and readability.

Reviewer #2

(Remarks to the Author)

Tian’s manuscript: “Identification of Thermotolerant Non-canonical PAMs for Robust One-Pot CRISPR-Cas12a Detection”

Tian et al. developed a poikilothermal, one-pot CRISPR-Cas12a detection platform (POP-CRISPR) by exploiting the PAM recognition properties of Cas12a. They showed that elevating the reaction temperature above 45 °C significantly enhanced the trans-cleavage activity of multiple non-canonical PAM sites, with some exhibiting activity comparable to that of canonical PAMs, while cis-cleavage activity remained relatively weak. Combining non-canonical PAMs with an elevated-temperature strategy further improved the system’s ability to discriminate mismatches, facilitating the identification of drug-resistant strains. POP-CRISPR outperformed canonical PAM-based methods in sensitivity, specificity, speed, and target versatility, and was successfully applied to rapid clinical detection of HPV and *Mycoplasma pneumoniae* within 20 minutes. The study highlights the potential of leveraging non-canonical PAMs together with temperature modulation to enhance Cas12a diagnostics. Nevertheless, the underlying mechanistic basis remains incompletely understood, and the data analysis and presentation warrant further elaboration and refinement.

Major:

1. Plot temperature-response curves for the trans-cleavage activity of each PAM to clearly show the optimal reaction temperature range.
2. Perform a quantitative analysis of the cis-cleavage efficiencies in Fig. 3 to allow a more accurate comparison across different temperatures.
3. Figure 4 shows that amplification at 37 °C for 10 minutes yields the best performance. Why not 30 minutes? Please clarify the kinetics of both amplification and cleavage.
4. The manuscript lacks a mechanistic analysis. Why does increasing the temperature selectively enhance trans-cleavage

activity without affecting cis-cleavage activity. Furthermore, why does a higher temperature reduce the system's tolerance to mismatches.

5. In Fig. 6, what are the Ct values of the 13 MP samples, and what is the limit of detection (LOD) of POP-CRISPR for clinical samples.

6. It would be important to clarify whether POP-CRISPR can also be applied to RNA samples, similar to the sPAMC.

7. The authors might consider introducing such mutations to enhance trans-cleavage activity, as exemplified by enAsCas12a (E174R/S542R/K548R), which retains activity at 60 °C.

Minor:

1. Please clarify the incubation conditions in Fig. 2, including whether any buffer was added during the pre-incubation/activation steps and whether Mg²⁺ was present (and at what concentration). Additionally, the Methods section should provide more detailed information on the experimental setup, including the exact composition of the Cas12a reaction buffer and the identity of 'polyC'—for example, whether it refers to a defined 6-mer (C₆) or a different length.

2. To better demonstrate the advantages of the proposed detection method, the authors should present a comparison table with qPCR.

3. The schematic in Fig. 6a is not sufficiently clear. It should be clarified whether the mini-device simultaneously performs both sample processing and detection reactions, or if it is limited to temperature control. The specific workflow and underlying mechanism should be explicitly described. We recommend including a more intuitive process illustration in the figure.

Version 1:

Reviewer comments:

Reviewer #1

(Remarks to the Author)

I have carefully reviewed the revised manuscript and the authors' detailed responses. The authors have addressed the major scientific concerns raised in the previous review. They have substantially strengthened the mechanistic explanations regarding the temperature-dependent behaviors of canonical and non-canonical PAMs, clarified the distinct activation mechanisms for dsDNA and ssDNA targets, and provided convincing new experimental evidence demonstrating that the observed thermal tolerance originates from the activated Cas12a ternary complex. In addition, the authors have directly tested both heat-tolerant and less heat-tolerant crRNAs, showing that the enhanced thermal robustness is independent of crRNA intrinsic stability and is instead determined by whether Cas12a has entered its target-activated state—thereby resolving the concern regarding the role of crRNA heat tolerance in the system's performance. All minor concerns have also been adequately addressed with appropriate clarifications or supplemental data.

Overall, I am satisfied that all issues have been resolved, and I believe the manuscript is now suitable for publication in Nature Communications.

Reviewer #2

(Remarks to the Author)

no more comments.

Point-by-Point Response to Reviewers

We thank both reviewers for their insightful comments. We respond to each of the points below and outline the changes we have made. We use black font for the referee comments, blue font for our responses, and *blue italic* font to reproduce the edits/addition we made to the manuscript/figures. In the attached “marked” manuscript all the changes are highlighted in red.

Reviewer: 1

Tian et al. systematically screened CRISPR-Cas12a PAM sites under elevated temperatures and identified over 80 non-canonical PAMs that exhibit trans-cleavage activities comparable to those mediated by canonical PAMs. In exploring the interplay between temperature and PAM identity in regulating Cas12a enzymatic activity and target specificity, the authors reported several notable findings that expand our understanding of the CRISPR-Cas12a system. For example, although trans-cleavage activity driven by non-canonical PAMs is markedly enhanced at elevated temperatures, the corresponding cis-cleavage remains relatively weak. This observation provides new evidence for the mechanistic decoupling of cis- and trans-cleavage activities in Cas12a. The authors further demonstrate that elevated temperature and non-canonical PAMs can act synergistically to improve target discrimination, achieving single-nucleotide resolution in SNP detection.

Building on these mechanistic insights, the authors developed a poikilothermal one-pot CRISPR platform based on thermotolerant non-canonical PAMs. This platform outperforms previous systems such as sPAMC in terms of sensitivity, specificity, and target flexibility. The applicability of POP-CRISPR for on-site detection was further demonstrated by optimizing sample processing protocols and integrating the assay with portable instrumentation. As such, this study integrates fundamental characterization of Cas12a biochemical behavior with the development of a practical diagnostic tool. Moreover, the synergistic enhancement of Cas12a activity and sequence specificity by temperature and PAM identity may be generalizable to other effectors in the Cas12 family, including Cas12b and Cas12f. Overall, the findings represent a meaningful contribution to both CRISPR biology and its diagnostic applications.

This is an interesting and valuable work, I strongly support publication of this manuscript in Nature Communications after addressing the following issues.

Response: We sincerely thank the reviewer for the positive and encouraging comments on our work. We greatly appreciate the time and effort you have invested in reviewing our work. We have carefully provided detailed, point-by-point responses to all the comments, and we hope that our revisions meet with your approval.

Major questions:

1. One of the key findings of this work is that the *trans*-cleavage activity of Cas12a is significantly enhanced at elevated temperatures, leading to the identification of over

80 previously unreported non-canonical PAMs with efficiencies comparable to the canonical ones. However, I noticed that the enhancement in *trans*-cleavage efficiency with increasing temperature appears to be more pronounced for crRNAs targeting non-canonical PAMs than for those targeting canonical PAMs (e.g., Figure 1). What might be the underlying reason for this difference? I believe a discussion of this point would not only strengthen the rigor of the conclusions but also enhance the reader's understanding of the work.

Response:

Thank you for this insightful comment. The more pronounced temperature-dependent enhancement observed for non-canonical PAM sites likely due to mechanistic differences in PAM affinity and R-loop activation, as detailed below.

First, non-canonical PAMs generally exhibit weaker binding affinity to Cas12a compared with canonical PAMs (Yamano et al., *Mol Cell*, 2017). At 37 °C, efficient R-loop formation at these sites requires greater local DNA breathing and transient base-pair melting to initiate target recognition. As the reaction temperature increases, DNA duplex unwinding becomes more frequent and energetically favorable, thereby lowering the activation barrier for R-loop formation at non-canonical PAM sites. In contrast, canonical PAMs provide strong and stable initial recognition even at 37 °C, so temperature elevation offers only limited additional benefit.

Second, PAM recognition triggers conformational rearrangements within Cas12a that are essential for activating its nuclease function (Stella et al., *Cell*, 2017). Structural studies have shown that a key residue (Lys595 in LbCas12a) inserts into the PAM minor groove to stabilize the complex. In non-canonical PAMs, this interaction is weakened due to suboptimal base pair geometry, often rendering the complex metastable at lower temperatures. Elevated temperatures may enhance the conformational flexibility of the Cas12a–DNA complex, reducing structural constraints between the PI domain and PAM duplex, and thereby facilitating transition into an active state.

Taken together, elevated temperature exerts limited influence on canonical PAM-mediated reactions but markedly improves recognition and conformational activation at non-canonical PAM sites.

We have incorporated this discussion into the revised manuscript (Page 4, Lines 122-125).

“We speculate that elevated temperature facilitates local DNA unwinding and enhances the conformational flexibility of the Cas12a complex, thereby lowering the activation barrier for non-canonical PAM recognition and cleavage. In contrast, canonical PAMs possess stronger intrinsic binding affinity and are thus less dependent on temperature elevation.”

2. In the experiments evaluating the temperature tolerance of different crRNAs (e.g., Figure 2c), the authors observed that for both canonical and non-canonical PAMs, the optimal reaction temperature was consistently around 45 °C, with some crRNAs

retaining considerable trans-cleavage activity even at temperatures as high as 57 °C. However, these experiments were all performed using double-stranded DNA (dsDNA) targets. Interestingly, when the targets were switched to corresponding single-stranded DNA (ssDNA), the optimal reaction temperature dropped to 37 °C, and increasing the temperature to 45 °C or higher led to a decrease in trans-cleavage activity (Supplementary Figure 7). What could be the mechanistic explanation for this discrepancy in temperature dependence between dsDNA and ssDNA targets? A discussion of this observation would enhance the understanding of the temperature-dependent behavior of Cas12a.

Response:

We sincerely thank the reviewer for this insightful question. In our study, LbCas12a exhibited distinct temperature dependencies when targeting dsDNA versus ssDNA substrates. With dsDNA targets, the optimal *trans*-cleavage activity was observed around 45 °C or even higher, whereas for ssDNA targets, the activity peaked at 37 °C and declined rapidly at elevated temperatures. We attribute this discrepancy to differences in Cas12a activation mechanisms.

Previous studies have shown that during dsDNA recognition, Cas12a first binds the PAM site and locally unwinds the DNA duplex to form an R-loop. When the crRNA–target strand pairing reaches ~18 bp, the R-loop achieves a stable checkpoint conformation that is critical for *trans*-cleavage activation (*Nature chemical biology*, 2022, 18(9), 1014-1022). We speculate that moderate temperature elevation facilitates DNA unwinding and R-loop formation, thereby enhancing Cas12a activation. In contrast, for ssDNA targets, activation no longer depends on R-loop formation but solely on crRNA–ssDNA hybridization. Higher temperatures can destabilize this pairing and promote complex dissociation, leading to reduced *trans*-cleavage activity.

We therefore speculate that the distinct optimal temperatures reflect these mechanistic differences: dsDNA-mediated activation benefits from temperature-enhanced DNA breathing and conformational flexibility, whereas ssDNA-mediated activation is limited by the thermal instability of the crRNA–target duplex.

We have incorporated a discussion of this mechanism in the revised manuscript (Page 6, Lines 167-174).

“To better understand this temperature dependence, we next examined target-type effects. Notably, this high-temperature tolerance mechanism appears to be more applicable to double-stranded DNA. When testing crRNAs with single-stranded targets, we observed that the highest activity was typically achieved at 37°C (Supplementary Fig. 7). This difference likely arises from distinct activation mechanisms: elevated temperature promotes R-loop formation and activation with dsDNA targets, whereas ssDNA activation depends solely on crRNA–target hybridization, which becomes destabilized at higher temperatures. Given this thermal sensitivity, we speculate that crRNA degradation may also contribute to activity loss at high temperatures.”

3. The authors mention that monomeric Cas12a protein is thermosensitive and can be nearly inactivated after 30 minutes of incubation at 37 °C. However, once activated by the target, the Cas12a RNP complex retains robust *trans*-cleavage activity even at 53 °C. I have two questions regarding this observation:

Comment 1: Was the crRNA used in this experiment itself one of the heat-tolerant crRNAs identified earlier? If a less heat-tolerant crRNA had been used, would the results have been different?

Response:

We sincerely appreciate the reviewer’s insightful question. In the thermal stability experiments of the LbCas12a reaction system (Figure 2d–g), the crRNA used was crRNA10 (as shown in Figure 2c), which exhibited strong *trans*-cleavage activity across a wide temperature range (37–57 °C) and was classified as a heat-tolerant crRNA.

To address the reviewer’s concern, we additionally tested two crRNAs—crRNA6 and crRNA12—whose *trans*-cleavage activities gradually decreased at higher temperatures. As shown in the new data below (Commentary Fig.1), results were consistent with those obtained using crRNA10: crRNA alone remained functional after incubation at various temperatures, and the LbCas12a–crRNA binary complex retained activity at 37 °C but became rapidly inactivated at elevated temperatures. In contrast, once Cas12a was activated by the target, the resulting ternary complex displayed robust *trans*-cleavage activity even at 53 °C.

These findings demonstrate that the thermal tolerance of the Cas12a reaction system is determined by the activation state of Cas12a rather than by the intrinsic heat tolerance of the crRNA itself.

Commentary Fig.1 Sensitivity assessment of the wild-type unmodified crRNA-mediated one-pot RPA-LbCas12a system. a Evaluation of the thermal stability of crRNA crRNA6 and

crRNA12). CrRNA was pre-incubated at 37°C, 45°C, or 53°C for 30 minutes, after which the remaining components (LbCas12a, dsDNA target and reporter) were added for the 45°C reaction and real-time fluorescence acquisition. Real-time fluorescence curve for sensitivity detection using wild-type unmodified crRNA. **b** Evaluation of the thermal stability of LbCas12a-crRNA RNP. RNP was pre-incubated at 37°C, 45°C, or 53°C for 30 minutes, after which the remaining components (dsDNA target and reporter) were added for the 45°C reaction and real-time fluorescence acquisition. **c** Evaluation of the thermal stability of dsDNA target activated LbCas12a. LbCas12a-crRNA RNP and target DNA were pre-incubated at 37°C, 45°C, or 53°C for 30 minutes, after which the reporter was added for the 45°C reaction and real-time fluorescence acquisition. ① and ② represent the order of addition. Data are represented as mean \pm standard error (n=3 technical replicates).

Comment 2: What might be the underlying reason for the improved thermal stability of target-activated RNPs compared to the monomeric Cas12a protein? Could this be due to conformational stabilization upon target binding? I suggest that the authors consider including more direct experimental evidence—such as differential scanning fluorimetry (DSF)—to further explore this aspect.

Response:

We sincerely thank the reviewer for this valuable and professional suggestion. Following your advice, we performed differential scanning fluorimetry (DSF) to examine the thermal stability of LbCas12a under different complex states. We selected two representative crRNAs from Figure 2c—crRNA 1 and crRNA 5—of which crRNA 1 exhibited robust *trans*-cleavage activity across 37–57 °C, whereas crRNA 5 showed approximately 75% loss of activity above 55 °C.

DSF analyses were conducted for monomeric LbCas12a, LbCas12a–crRNA (binary) complexes, and LbCas12a–crRNA–dsDNA (ternary) complexes. The results showed no significant difference in melting temperature (T_m) among these forms, all displaying $T_m \approx 44.8$ °C (Commentary Fig.2). This indicates that the overall thermal stability of LbCas12a does not markedly change upon crRNA binding or target activation, consistent with previous observations (*Cell Reports*, 2024, 43(2)).

Although no global T_m shift was observed, we speculate that this does not exclude the possibility of local conformational stabilization upon target binding. Structural studies have demonstrated that, upon crRNA–target DNA duplex formation, Cas12a undergoes a major conformational transition from an open, flexible state to a tightly closed configuration. This transition reinforces interactions among the REC and NUC lobes and the RuvC catalytic center (*Molecular Cell*, 2024, 84(14): 2717–2731), thereby reducing interdomain mobility and potentially enhancing the functional thermal tolerance of the activated complex.

In addition, the R-loop formed between crRNA and the target DNA may provide further stabilization through additional hydrogen bonding and base-stacking interactions, making the ternary complex more resistant to thermal dissociation. We therefore hypothesize that the improved thermal stability of target-activated Cas12a primarily reflects local conformational stabilization and R-loop-mediated clamping

effects, which are not readily captured by bulk unfolding measurements such as DSF. Further investigations using higher-resolution structural or dynamic techniques—such as single-molecule FRET or cryo-EM—may be required to directly validate this proposed mechanism.

Commentary Fig.2 Differential scanning fluorimetry (DSF) derivative plots. DSF analyses were performed for monomeric LbCas12a, LbCas12a–crRNA binary complexes, and LbCas12a–crRNA–dsDNA ternary complexes. The global minima of the derivative curves were taken as the melting temperatures (T_m) of the respective proteins or complexes.

4. In Lines 226–238, the authors report an interesting observation: experimental results demonstrate that increasing the reaction temperature enhances the *trans*-cleavage activity of Cas12a to the extent that it can even degrade blunt-ended double-stranded DNA—a phenomenon not previously reported. However, this section only describes the experimental outcome without offering any discussion or analysis of the underlying mechanism. I recommend that the authors include an appropriate and objective discussion on the possible reasons why Cas12a is capable of efficiently degrading a broader range of *trans*-cleavage substrates at elevated temperatures.

Response:

We sincerely thank the reviewer for this constructive comment. Early studies generally suggested that Cas12a lacks *trans*-cleavage activity toward double-stranded DNA (dsDNA) substrates because the narrow RuvC catalytic pocket cannot accommodate duplex DNA (*Science*, 2018, 360, 436-439). However, subsequent reports have shown that Cas12a can degrade dsDNA containing local single-stranded features, such as bulged regions along λ -DNA backbones or dsDNA with 3' overhangs (*Nucleic Acids Research*, 2023, 51, 9894-9904). In these cases, Cas12a essentially targets the transient single-stranded regions, and the apparent dsDNA degradation results from cleavage of these locally unpaired segments. By contrast, blunt-ended dsDNA—which lacks such single-stranded features—has been considered resistant to *trans*-cleavage.

Based on this understanding, we speculate that the phenomenon observed in our study is primarily attributable to temperature-induced alterations in DNA conformational dynamics. As the reaction temperature increases from 37 °C to 45 °C or higher, the rate of DNA “breathing” and local strand separation substantially rises (*Biophysical journal*, 2007, 92(8), 2674-2684), transiently exposing additional single-stranded regions within the duplex, which serve as accessible sites for Cas12a *trans*-cleavage. These fleetingly unpaired segments could serve as accessible sites for

Cas12a-mediated *trans*-cleavage, thereby enabling efficient degradation of even blunt-ended dsDNA substrates at elevated temperatures.

This finding not only broadens the known substrate scope of Cas12a *trans*-cleavage but also underscores that its enzymatic behavior is strongly influenced by the conformational dynamics of the DNA substrate, particularly under thermal modulation.

The relevant discussion has been incorporated into the revised manuscript (Page 8, Lines 271-275).

“We speculate that this phenomenon results from enhanced DNA “breathing” at elevated temperatures, which transiently exposes single-stranded regions accessible to Cas12a’s RuvC domain. Consequently, higher temperature increases the availability of cleavable single-stranded sites within dsDNA, enabling Cas12a to degrade a broader range of trans-cleavage substrates.”

Minor questions:

1. How were the scatter plot data in Figures 1c and 1d generated? The authors should clarify how these data were calculated or derived. A brief description in the figure legend or the Methods section would enhance reproducibility.

Response:

We sincerely thank the reviewer for this insightful comment. The data processing method for the scatter plots in Figures 1c and 1d followed the approach described in *Nature Biomedical Engineering* 6.3 (2022): 286–297. In these plots, each data point represents a specific PAM site: the y-axis denotes the average fluorescence intensity (defined as the maximum fluorescence value) obtained from three replicate reactions at 30 minutes, while the x-axis represents the time required for the fluorescence signal to reach half of that maximum value.

In the revised manuscript, we have added a brief description of this analysis to the legend of Figure 1 for clarity and reproducibility (Page 5, Lines 140-143).

“In these plots, each data point represents a specific PAM site: the y-axis denotes the average fluorescence intensity (defined as the maximum fluorescence value) obtained from three replicate reactions at 30 minutes, while the x-axis represents the time required for the fluorescence signal to reach half of that maximum value.”

2. In Figure 1, the PAM screening was performed using a ASFV P72 gene fragment, whereas Figures 4a–g focus on the detection of HPV16 and HPV18. Were the non-canonical PAMs used in these assays (such as CCCC and CCTG) also experimentally validated with the corresponding HPV targets? More generally, are the over 80 identified suboptimal non-canonical PAMs broadly applicable across different target sequences, or is their effectiveness sequence-dependent?

Response:

We sincerely appreciate the reviewer’s insightful question. In the comprehensive PAM screening experiment shown in Figure 1, a 296 bp fragment of the ASFV P72 gene was used as the target, and all PAM variants were evaluated using the same

spacer sequence of a single crRNA. Under the 45 °C condition, we identified more than 80 suboptimal non-canonical PAMs that exhibited *trans*-cleavage activities comparable to those of canonical PAMs.

To further verify whether these non-canonical PAMs are applicable to different target sequences, we conducted additional experiments using multiple crRNAs designed against distinct regions of the SARS-CoV-2 S gene, each associated with various suboptimal PAMs (Figure 2). The results (Figure 2c) showed that at 45 °C, the *trans*-cleavage activities of these crRNAs were consistently and markedly enhanced relative to those at 37 °C. Furthermore, prior to the one-pot detection assays in Figure 4, we independently validated the activity of each non-canonical PAM employed, including CCCC for HPV16, CCTG for HPV18, and TTAG for *Mycoplasma pneumoniae* (MP) (Commentary Fig.3). Comparative analyses revealed that all these crRNAs exhibited significantly elevated *trans*-cleavage activities under the elevated temperature condition. These validation experiments had been performed during the initial study but were not included in the original submission. We have now added these previously obtained results in response to the reviewer's request.

Collectively, these results demonstrate that the identified suboptimal non-canonical PAMs are not restricted to a specific target sequence but are generally applicable across diverse loci, with their enhanced activity primarily attributed to temperature-dependent regulation rather than strict sequence context.

Commentary Fig.3 Real-time fluorescence kinetics of three non-canonical PAM crRNAs (used in the one-pot assay shown in Figure 4) at 37 °C and 45 °C. a Real-time fluorescence kinetics of HPV-16 CCCC-crRNA at 37 °C and 45 °C. **b** Real-time fluorescence kinetics of HPV-18 CCTG-crRNA at 37 °C and 45 °C. **c** Real-time fluorescence kinetics of MP TTAG-crRNA at 37 °C and 45 °C. A target final concentration of 100 pM was used. Data are represented as mean \pm standard error (n=3 technical replicates).

3. Although the scatter plots in Figure 1 show that the number of non-canonical PAMs with increased reaction rates grows substantially with rising temperature, this presentation only provides an overall view of the data and does not link the results to specific PAM sites. To facilitate future reference and selection of suitable non-canonical PAMs by readers, I suggest that the authors consider including the fluorescence reaction rate data corresponding to individual PAM sites in the supplementary materials.

Response:

We sincerely appreciate your valuable and constructive suggestion. The scatter plots in Figure 1 were originally designed to provide an overall visualization of the global trend that the number of suboptimal non-canonical PAMs with enhanced

activity markedly increases with elevated temperature. In accordance with your recommendation, we have now performed a comprehensive statistical analysis of the reaction rates corresponding to all 256 PAM sequences and plotted the results as bar graphs, in which each bar represents an individual PAM site. The complete dataset has been included in the Supplementary Materials to enable readers to more accurately and conveniently reference specific non-canonical PAMs and to facilitate the rational selection of suitable PAM sequences in future applications. Furthermore, to ensure data completeness and transparency, the original real-time fluorescence acquisition data for all PAM sites at both 37 °C and 45 °C have been provided in the source data file, allowing interested readers to further examine and analyze the raw experimental results.

Supplementary Fig.2 | Fluorescence kinetics of 256 PAMs at 37 °C and 45 °C . a-d Fluorescence kinetics of 256 PAMs were measured in collateral activity experiments at 37°C. **e-h** Fluorescence kinetics of 256 PAMs were measured in collateral activity experiments at 45°C. The slope represents the rate of fluorescence change during the first four minutes, prior to reaching the maximum fluorescence value. The concentrations of dsDNA targets were 100 pM. Data are represented as mean ± standard error (n=3 technical replicates).

4. When assessing the effect of temperature on Cas12a enzymatic activity (e.g., in Figure 2c), the authors selected 37, 45, 53, 55, and 57°C. What was the rationale for selecting these specific temperature points? The author should explain the reasoning for selecting these specific temperature values. Was this based on enzyme kinetics, thermostability, or empirical observations?

Response:

We sincerely appreciate the reviewer’s insightful comments. The selection of

temperature points was guided by both established knowledge of Cas12a enzymatic behavior and our empirical optimization results. Specifically, 37 °C is generally recognized as the optimal reaction temperature for LbCas12a and was therefore chosen as the baseline control. Our preliminary experiments revealed that the *trans*-cleavage activity of Cas12a markedly increases at moderately elevated temperatures, with a pronounced enhancement observed around 45 °C. Hence, this temperature was included as a representative intermediate condition. To maintain a consistent thermal gradient and to capture potential transition points in enzyme activity, we further selected 53 °C. Given the likelihood of rapid thermal inactivation at higher temperatures, 55 °C and 57 °C were incorporated with narrower intervals to finely probe the upper thermal limit of Cas12a activity. Collectively, this temperature series was designed to systematically delineate the relationship between enzymatic activity and thermal stability, integrating both empirical observations and theoretical considerations of Cas12a kinetics.

5. I noticed that in the experiments assessing Cas12a *trans*-cleavage activity at different temperatures, the concentration of the Cas12a–crRNA RNP complex was consistently set at 10 nM. I am curious whether increasing the RNP concentration might further improve the thermal tolerance of the system. I suggest the authors explore this possibility, as it could provide useful insights into optimizing reaction conditions under elevated temperatures.

Response:

We sincerely appreciate your insightful comment. Following your suggestion, we increased the concentration of the Cas12a–crRNA RNP complex from 10 nM to 100 nM while maintaining the target DNA at 100 pM. We then evaluated the *trans*-cleavage activity of crRNAs 1-15 across temperatures ranging from 37 °C to 57°C (Figure 2c). The results showed that when the RNP-to-target ratio increased from 100:1 to 1000:1, excessively high RNP concentrations actually had a negative effect on *trans*-cleavage activity at all tested temperatures (Commentary Fig.4). We speculate that this may be because the large excess of inactive RNP molecules in the reaction mixture could compete with the activated RNPs for binding to the ssDNA reporter, thereby hindering efficient cleavage.

Commentary Fig.4 Heatmaps of trans-cleavage reaction rates for varying concentrations of Cas12a–crRNA RNP complexes across different reaction temperatures. Each heatmap represents the average reaction rate measured for canonical PAM crRNAs 1–15. The slope represents the rate of fluorescence change during the first four minutes, prior to reaching the maximum fluorescence value. The concentrations of S-gene dsDNA targets were 100 pM. Data are represented as mean \pm standard error (n=3 technical replicates).

6. In lines 214–216, the authors state that: “...the *cis*- and *trans*-cleavage activities of Cas12a are independent of each other.” A recent preprint (bioRxiv 2025.03.23.644851; doi: <https://doi.org/10.1101/2025.03.23.644851>) reports similar findings. The authors are encouraged to cite this reference to support their claim.

Response:

We sincerely thank the reviewer for the valuable suggestion. We note that this preprint has now been published in *Nucleic Acids Research*, and we have added the corresponding citation in the revised manuscript to support our conclusion.

7. In Figures 4h to 4k, would it be possible to add individual data points to the bar plots? Doing so would allow readers to better assess the variability across replicates. The same issue is also present in Supplementary Figures 6 to 8, and I recommend that the authors consider making this adjustment consistently across all relevant figures.

Response:

We sincerely appreciate the reviewer’s helpful suggestion. In accordance with your recommendation, we have redrawn all bar plots involving replicate experiments in both the main text and Supplementary Materials. Individual data points from the three independent replicates have been added to each bar to more clearly represent data distribution and experimental variability.

8. Figures 4h and 4j include double-base substitutions across all 20 spacer positions,

but Figures 4i and 4k only show selected single-nucleotide variants. How were the positions for single-nucleotide substitutions chosen, and could additional explanation be given?

Response:

We sincerely thank the reviewer for this insightful comment. The selection of single-nucleotide substitution sites in Figures 4i and 4k was not random but was guided by the results of the double-base substitution screening performed at 45 °C (Figures 4h and 4j). In those experiments, we observed that substitutions at certain positions (e.g., positions 3–4 and 9–16) caused a marked reduction or complete loss of Cas12a *trans*-cleavage activity, indicating high mismatch sensitivity at these loci. We therefore hypothesized that even single-nucleotide mismatches at these sensitive positions could significantly affect Cas12a activity. Conversely, positions showing tolerance to double substitutions (e.g., 1–2, 5–6, and 17–20) were expected to be less affected by single mismatches. Based on this rationale, we selectively introduced single-nucleotide substitutions at the mismatch-sensitive positions to more efficiently and mechanistically elucidate the influence of temperature and PAM context on Cas12a mismatch tolerance, rather than performing single-nucleotide substitution analysis across all 20 spacer positions.

9. The authors employed a Chelex-100-based thermal lysis protocol for nasopharyngeal swabs (Figures 6b–c). Could this method be extended to other sample types such as cervical swabs or blood-derived fluids?

Response:

We sincerely appreciate your insightful comment. Chelex-100 is a heat-stable phenolic chelating resin widely used for rapid and low-cost DNA extraction. In Figures 6b–c, we applied a Chelex-100-based thermal lysis method to process nasopharyngeal swab samples for rapid nucleic acid release compatible with downstream POP-CRISPR detection.

To evaluate the broader applicability of this approach, we further tested it on clinical cervical swab samples and compared the results with those obtained using a standard nucleic acid extraction kit. The results showed that nucleic acids extracted using the Chelex-100 thermal lysis method performed comparably in both qPCR and POP-CRISPR assays (Commentary Fig. 5), demonstrating that this simple approach maintains sufficient analytical sensitivity and specificity.

Moreover, previous studies have reported the successful use of Chelex-100 for rapid nucleic acid extraction from a wide range of sample types—including serum, whole blood, dried blood spots (*PNAS*, 2020, 25722–25731), lesion exudate swabs, oral swabs, saliva, and urine (*Nature Communications*, 2024, 3279)—followed by CRISPR-based detection. These findings further support the versatility and general applicability of Chelex-100 across various clinical sample types.

Commentary Fig. 5 Real-time fluorescence kinetics of clinical cervical swab samples. a Real-time fluorescence curves of qPCR using clinical cervical swab samples processed with different nucleic acid extraction methods. **b** Real-time fluorescence curves of POP-CRISPR using clinical cervical swab samples processed with different nucleic acid extraction methods.

10. There are a few typographical and formatting issues that merit attention. For example, the term “trans” in line 119 should be properly formatted, and the label in Figure 5a should read “RAA-CRISPR/Cas12a” instead of “RRA-CRISPR/Cas12a”.

Response:

We sincerely appreciate your careful review and thank you for pointing out these typographical and formatting issues. We apologize for the oversights and have corrected all errors, including the formatting of “*trans*” in line 119 and the label in Figure 5a, which now correctly reads “RAA-CRISPR/Cas12a.”

11. In Supplementary Table S5, the lack of dividing lines makes it difficult to distinguish between the two plasmid sequences. I recommend that the authors carefully check and reformat the table to clearly separate the sequences for clarity and readability.

Response:

We sincerely appreciate your suggestion. The table has been carefully reformatted to include dividing lines, clearly separating the two plasmid sequences for improved clarity and readability.

Reviewer: 2

Tian et al. developed a poikilothermal, one-pot CRISPR-Cas12a detection platform (POP-CRISPR) by exploiting the PAM recognition properties of Cas12a. They showed that elevating the reaction temperature above 45 °C significantly enhanced the trans-cleavage activity of multiple non-canonical PAM sites, with some exhibiting activity comparable to that of canonical PAMs, while cis-cleavage activity remained relatively weak. Combining non-canonical PAMs with an elevated-temperature strategy further improved the system's ability to discriminate mismatches, facilitating the identification of drug-resistant strains. POP-CRISPR outperformed canonical PAM-based methods in sensitivity, specificity, speed, and target versatility, and was successfully applied to rapid clinical detection of HPV and *Mycoplasma pneumoniae* within 20 minutes. The study highlights the potential of leveraging non-canonical PAMs together with temperature modulation to enhance Cas12a diagnostics. Nevertheless, the underlying mechanistic basis remains incompletely understood, and the data analysis and presentation warrant further elaboration and refinement.

Response:

We sincerely thank the reviewer for the careful evaluation of our manuscript and for the constructive comments and suggestions. We greatly appreciate the time and effort you have invested in reviewing our work. We believe these valuable insights have helped us to further improve the clarity and quality of the paper. We have carefully addressed each point in detail, and we hope the revisions meet your expectations.

Major questions:

1. Plot temperature-response curves for the *trans*-cleavage activity of each PAM to clearly show the optimal reaction temperature range.

Response:

We sincerely thank the reviewer for this valuable suggestion. The scatter plots in Figure 1 were designed to illustrate the overall trend that the number of suboptimal non-canonical PAMs with detectable activity markedly increases with rising temperature. Following the reviewer's advice, we have now quantified the reaction rates for all 256 PAM sites and presented the results as bar charts, with each bar corresponding to an individual PAM site. These data have been included in the Supplementary Materials to facilitate clear and intuitive reference for readers when selecting suitable non-canonical PAMs.

It should be noted that since 256 PAMs were analyzed in total, presenting all raw fluorescence curves would result in an excessively large dataset that is difficult to compare visually. Therefore, we believe the bar-chart representation provides a clearer and more concise summary of the key findings. To ensure data completeness, we have also included the original real-time fluorescence data for all PAM sites under both 37 °C and 45 °C conditions in the source dataset, allowing interested readers to further examine and analyze the results.

Supplementary Fig.2 | Fluorescence kinetics of 256 PAMs at 37°C and 45°C. a-d Fluorescence kinetics of 256 PAMs were measured in collateral activity experiments at 37°C. e-h Fluorescence kinetics of 256 PAMs were measured in collateral activity experiments at 45°C. The slope represents the rate of fluorescence change during the first four minutes, prior to reaching the maximum fluorescence value. The concentrations of dsDNA targets were 100 pM. Data are represented as mean \pm standard error (n=3 technical replicates).

2. Perform a quantitative analysis of the *cis*-cleavage efficiencies in Fig. 3 to allow a more accurate comparison across different temperatures.

Response:

We sincerely thank the reviewer for this valuable comment. We analyzed the gel electrophoresis bands shown in Figure 3a–d using ImageJ to quantitatively assess the *cis*-cleavage efficiency of Cas12a under different PAM and temperature conditions. Specifically, the unreacted target DNA lane was used as a reference, and the grayscale ratio between uncleaved targets and the reference band was calculated to obtain quantitative *cis*-cleavage efficiency. Based on these results, we generated heatmaps illustrating the *cis*-cleavage efficiencies of different PAMs across various temperatures (Fig. 3e–h). The analysis revealed that Cas12a could mediate *cis*-cleavage within the 37–55 °C range, However, canonical PAMs (e.g., TTTA) exhibited substantially higher cleavage efficiencies than non-canonical PAMs (e.g., GTTA and CTTA). Increasing the temperature (≥ 45 °C) did not further enhance *cis*-cleavage activity.

In addition, we observed slight band shifts or smearing under prolonged incubation or elevated temperature conditions (≥ 45 °C). We speculate that this phenomenon may

PAM. **c, d** In vitro dsDNA cleavage activities of LbCas12a mediated by non-canonical PAM (TCCA, TTGG). These four crRNAs and the dsDNA target used are from **Fig.2a-b**. Double-stranded DNA cleavage activities were analyzed using 2% agarose gels at 10, 20, 30, 60 or 90 minutes across different temperatures (37-55°C). The final concentration of dsDNA used was 120 nM. **e-h** Quantitative analysis of Cas12a *cis*-cleavage efficiencies under different temperatures. *Cis*-cleavage efficiencies were calculated from the electrophoretic bands in Fig.3a-d by densitometric analysis using Image J. For each lane, the efficiency was determined as the difference between the reference lane (unreacted control) and the uncleaved target band, divided by the reference lane intensity. Heatmaps display the temperature-dependent *cis*-cleavage efficiencies of canonical PAMs (**e, f**) and non-canonical PAMs (**g, h**). **i** Schematic diagram of diverse substrates for *trans*-cleavage with activated Cas12a. When Cas12a is activated by target (ssDNA/dsDNA), it can *trans*-cleave dsDNA target, *cis*-cleaved DNA substrates, any double-stranded DNA, and any single-stranded DNA substrates. Additionally, this *trans*-cleaving activity increases with rising temperature. Source data are provided as a Source Data file.

3. Figure 4 shows that amplification at 37 °C for 10 minutes yields the best performance. Why not 30 minutes? Please clarify the kinetics of both amplification and cleavage.

Response:

Thank you very much for your valuable comment. In the POP-CRISPR system, we employed a suboptimal PAM-mediated crRNA. During the initial stage of the reaction at 37 °C, Cas12a exhibits relatively slow cleavage kinetics toward the target or amplification products, which allows the reaction equilibrium to favor isothermal amplification and thereby rapidly accumulate amplicons. When the temperature is subsequently raised to 45 °C, Cas12a reaches its optimal activity range, exhibiting a markedly enhanced *trans*-cleavage efficiency that degrades the reporter probes and generates detectable fluorescence signals.

In Figure 4e–f, we compared one-pot assays pre-amplified at 37 °C for 5, 10, and 30 minutes. All three conditions produced positive signals that were clearly distinguishable 10 aM target from the NTC. However, the 10-minute condition yielded the earliest onset and the highest terminal fluorescence signal. To further examine the amplification kinetics, we analyzed the products after different pre-amplification durations by agarose gel electrophoresis. The results showed that amplification bands were barely visible at 5 minutes, became clearly detectable at 10 minutes, and showed no significant increase at 30 minutes (Supplementary Fig.18). These findings indicate that amplification was nearly saturated at approximately 10 minutes, and extending the reaction to 30 minutes did not further increase the product yield.

Therefore, the higher fluorescence observed at the 10-minute condition does not result from greater amplicon accumulation but rather from the earlier transition to 45 °C, which triggers the highly efficient Cas12a *trans*-cleavage phase sooner, leading to an earlier and stronger overall signal. Taken together, a 10-minute pre-amplification at 37 °C is sufficient to achieve high-sensitivity detection while avoiding signal delay caused by prolonged amplification, representing the optimal reaction condition for the

POP-CRISPR assay.

The relevant experimental results and discussion have been incorporated into the revised manuscript (Page 10, Lines 320-325).

“Electrophoretic analysis further confirmed that amplicon accumulation nearly reached saturation after 10 minutes, with no significant increase observed at 30 minutes (Supplementary Fig. 18). Thus, for the POP-CRISPR method, a 10-minute pre-amplification at 37 °C is sufficient to support high-sensitivity detection without causing signal delay, as prolonged amplification primarily postponed the initiation of highly efficient Cas12a trans-cleavage activity (Fig. 4g).”

Supplementary Fig.18 | The accumulation of RAA amplicons in the POP-CRISPR system. a Reaction mixtures containing 10 nM Cas12a/crRNA RNP and 10 aM HPV-16 dsDNA substrates were incubated at 37 °C for 5, 10, and 30 min. **b** Reaction mixtures containing 10 nM Cas12a/crRNA RNP and 10 aM HPV-18 dsDNA substrates were incubated at 37 °C for 5, 10, and 30 min. The resulting RAA amplicons from both reactions were analyzed in 2% agarose gels.

4. The manuscript lacks a mechanistic analysis.

Comment 1: Why does increasing the temperature selectively enhance *trans*-cleavage activity without affecting *cis*-cleavage activity.

Response:

We sincerely thank the reviewer for raising this important question. We agree that distinguishing the effects of temperature on *cis*- versus *trans*-cleavage activities of Cas12a and discussing the underlying mechanisms is critical.

Structural studies of Cas12a have shown that upon target recognition, the protein undergoes significant conformational changes, transitioning from a relatively flexible open state to a tightly bound closed state (*Molecular Cell*, 2024, 84(14): 2717–2731). *Cis*-cleavage occurs once the crRNA–DNA hybrid forms an R-loop, directing the target strand precisely into the RuvC catalytic pocket. This process is largely determined by sequence-specific crRNA–DNA interactions and local structural positioning, making it relatively insensitive to moderate temperature changes.

In contrast, *trans*-cleavage requires further conformational rearrangements of Cas12a following initial *cis*-cleavage and R-loop stabilization, enabling the RuvC catalytic site to repeatedly engage freely diffusing, non-target single-stranded nucleic acids. Elevating the reaction temperature likely increases the diffusion and effective collision frequency of these *trans* substrates, thereby enhancing *trans*-cleavage efficiency.

Taken together, these considerations suggest that the selective enhancement of *trans*-cleavage at elevated temperatures, without a corresponding increase in *cis*-cleavage, primarily reflects mechanistic differences between the two activities: *cis*-cleavage is constrained by sequence-specific recognition and R-loop formation, whereas *trans*-cleavage is more dependent on substrate diffusion and access to the catalytic pocket.

The relevant discussion has been incorporated into the revised manuscript (Page 8, Lines 242-253).

“Together with the earlier trans-cleavage results, these observations indicate that elevating the reaction temperature predominantly enhances trans-cleavage activity. This effect likely reflects the intrinsic mechanistic differences between the two catalytic modes of Cas12a. During cis-cleavage, target recognition, PAM engagement, and R-loop formation precisely orient the target strand into the RuvC catalytic pocket, making this process largely dictated by stable crRNA–DNA base pairing and structural positioning and therefore relatively insensitive to moderate temperature elevation. By contrast, trans-cleavage requires additional conformational rearrangements following R-loop stabilization to permit repeated access of freely diffusing single-stranded substrates to the active site. Increased temperature likely accelerates the diffusion and collision frequency of these trans substrates and facilitates their engagement with the catalytic pocket, thereby selectively boosting trans-cleavage efficiency without proportionally affecting cis-cleavage.”

Comment 2: Furthermore, why does a higher temperature reduce the system’s tolerance to mismatches.

Response:

Thank you for this insightful question. First, increasing reaction temperature reduces the thermodynamic stability of crRNA–target base pairing. This destabilization is accentuated at mismatch sites, where hydrogen bonds are more readily disrupted, thereby lowering Cas12a’s tolerance for mismatched substrates. Second, non-canonical PAMs exhibit intrinsically weaker binding affinity to Cas12a (*Molecular Cell*, 2017, 67(4):633–645.e3), which makes target recognition and subsequent cleavage more dependent on strict crRNA–target complementarity. The combination of these two factors further lowers the threshold for mismatch tolerance, such that only fully complementary duplexes reliably form active cleavage complexes — effectively enabling single-nucleotide discrimination. To illustrate this mechanism, we have added a schematic in the Supplementary Materials that depicts how Cas12a

recognition and cleavage of highly similar sequences differ under varying temperature conditions.

The relevant experimental results and discussion have been incorporated into the revised manuscript (Page 11, Lines 362-370)

“Mechanistically, elevated temperatures decrease the thermodynamic stability of crRNA–target hybridization, and this destabilization is disproportionately amplified at mismatched positions, where hydrogen bonding is more easily disrupted. As a result, imperfect duplexes fail to maintain the stable R-loop required for efficient Cas12a activation. In addition, non-canonical PAMs intrinsically exhibit weaker affinity for Cas12a, thereby increasing the reliance on stringent crRNA–target complementarity for productive target engagement and cleavage. The combination of reduced hybrid stability and lower PAM-binding strength at higher temperatures substantially narrows the tolerance window for mismatches, effectively enabling single-nucleotide discrimination and further improving detection specificity (Supplementary Fig. 20).”

Supplementary Fig.20 | Schematic illustration of the synergistic regulation of Cas12a detection specificity by temperature and non-canonical PAMs. At low temperatures (<45 °C), both perfectly matched and mismatched targets can activate Cas12a due to stable RNP binding. Increasing the temperature (>53 °C) destabilizes mismatched crRNA–DNA pairs, and the weaker affinity of non-canonical PAMs further limits Cas12a activation to fully matched targets, enabling single-base discrimination.

5. **Comment 1:** In Fig. 6, what are the Ct values of the 13 MP samples?

Response:

We have provided the Ct values of the 13 *Mycoplasma pneumoniae*–positive clinical samples used in Figure 6c. This information has been added to the revised Supplementary Materials.

Table S11 The Ct values of 27 clinical MP samples used in Fig.6 c-d measured by qRT-PCR.

Sample ID	1	2	3	4	5	6	7	8	9
MP (Ct)	24.44	32.48	33.79	36.01	34.93	31.16	25.71	22.41	23.82
Sample ID	10	11	12	13	14	15	16	17	18
MP (Ct)	24.77	32.87	33.97	21.99	-	-	-	-	-

Sample ID	19	20	21	22	23	24	25	26	27
MP (Ct)	-	-	-	-	-	-	-	-	-

Comment 2: What is the limit of detection (LOD) of POP-CRISPR for clinical samples.

Response:

Thank you for your question. In the analytical validation experiments using purified PCR amplicons as targets, the limit of detection (LOD) of POP-CRISPR was determined to be approximately 0.5 aM, corresponding to 4–5 copies per reaction (Fig. 4 and Supplementary Fig. 19). For clinical samples, POP-CRISPR successfully detected HPV-16 with qPCR Ct values up to 35.7 and MP clinical samples with qPCR Ct values up to 36.2, which are close to the detection threshold of standard qPCR assays. Therefore, while an exact copy number–based LOD cannot be directly assigned to clinical specimens, these results indicate that POP-CRISPR achieves a practical detection sensitivity comparable to its analytical LOD, enabling reliable detection of ultra-low-abundance targets in real clinical contexts.

Supplementary Fig.19 | Reproducibility of POP-CRISPR platform. **a** 20 experiments were repeated independently with HPV-16 DNA detection at a concentration of 9, 4.5, 1.8, and 0 copies/reaction. Threshold was determined by the average fluorescence values of 20 NTC plus 3 times of standard deviation. **b** Positive detection rates of detecting with different concentrations of HPV-16 DNA in (a). The volume of the reaction system was 15 μ L.

6. It would be important to clarify whether POP-CRISPR can also be applied to RNA samples, similar to the sPAMC.

Response:

We sincerely thank the reviewer for this valuable comment. We further evaluated the feasibility of applying the POP-CRISPR system to RNA targets. Specifically, we used *in vitro*-transcribed RNA fragments of the SARS-CoV-2 N gene as templates and compared the performance of POP-CRISPR with that of sPAMC. A crRNA designed with a non-canonical CCCA PAM was employed in both systems. The results showed that POP-CRISPR successfully detected RNA targets down to 1 aM, achieving a detection limit comparable to sPAMC, while exhibiting higher endpoint fluorescence intensity and faster signal accumulation. These results indicate that POP-

CRISPR is fully compatible with RNA detection and provides enhanced sensitivity and reaction kinetics relative to existing systems.

The relevant experimental results and discussion have been incorporated into the revised manuscript (Page 10, Lines 319-322)

“Furthermore, we validated the applicability of POP-CRISPR for RNA target detection using *in vitro*-transcribed SARS-CoV-2 N gene RNA, showing a detection limit of 1 aM with higher endpoint signal intensity and faster reaction kinetics than sPAMC (Supplementary Fig. 17).”

Supplementary Fig.17 | Evaluation of the sensitivity of POP-CRISPR and sPAMC on RNA target. **a** Real-time fluorescence signal curves for the detection of SARS-CoV-2 N gene by POP-CRISPR. **b** Endpoint fluorescence signals (60 min) for each RNA target concentration in panel **a**. **c** Real-time fluorescence signal curves for the detection of SARS-CoV-2 N gene by sPAMC. **d** Endpoint fluorescence signals (60 min) for each RNA target concentration in panel **c**. RNA targets were serially diluted with final concentrations of 10 fM, 1 fM, 100 aM, 10 aM, and 1 aM. All experiments were performed in triplicate, with error bars representing the mean values +/- SD (n = 3), and statistical analysis was conducted using a two-tailed t-test. Statistical significance is indicated as follows: n.s. = no significance with P > 0.05, and the asterisks (* P < 0.05; ** P < 0.01; *** P < 0.001; **** P < 0.0001.). a.u. represents arbitrary units.

7. The authors might consider introducing such mutations to enhance *trans*-cleavage activity, as exemplified by enAsCas12a (E174R/S542R/K548R), which retains activity at 60 °C.

Response:

We sincerely thank the reviewer for this valuable suggestion. Following the reviewer’s advice, we designed corresponding mutations in LbCas12a based on sequence alignment and structural comparison with enAsCas12a (E174R/S542R/K548R) (*Nature communications*, 2021, 12(1): 1739.). Three residues were identified as potential homologous sites (D156R, G532R, and K538R)

(Supplementary Fig.9), and the resulting mutant protein (LbCas12a-Mut) was successfully expressed and purified.

To evaluate whether these substitutions enhance the *trans*-cleavage activity of LbCas12a-Mut at elevated temperatures, we selected four representative crRNAs from Fig. 2c: crRNA9 and crRNA17 (temperature-sensitive, with *trans*-cleavage activity decreasing progressively above 53 °C and dropping by ~80% at 57 °C), and crRNA10 and crRNA20 (thermotolerant, retaining strong activity at 57 °C). We tested the *trans*-cleavage performance of LbCas12a-Mut under these crRNAs across 55–59 °C. The results showed that LbCas12a-Mut displayed a similar temperature-dependent pattern to the LbCas12a-WT (Supplementary Fig.10a). For temperature-sensitive crRNAs, *trans*-cleavage activity declined sharply above 53 °C and was completely lost at 59 °C. Thermotolerant crRNAs maintained robust activity at 57 °C but also became inactive at 59 °C. These findings indicate that the introduced mutations did not enhance the intrinsic thermostability or high-temperature *trans*-cleavage activity of LbCas12a.

In addition, we noted that the study reporting thermostable activity of enAsCas12a at 60 °C employed Tango buffer (*Nature communications*, 2021, 12(1): 1739.). To examine whether buffer composition influences thermostability, we repeated the above assays using Tango buffer. Strikingly, LbCas12a-Mut exhibited markedly improved performance under this condition (Supplementary Fig.10b). For temperature-sensitive crRNAs, approximately 80% activity was retained at 57 °C, while thermotolerant crRNAs showed even stronger enhancement, maintaining substantial activity at 59 °C. To further determine the upper temperature limit of this buffer-mediated thermostability, we next evaluated both LbCas12a-WT and LbCas12a-Mut using the thermotolerant crRNAs (Supplementary Fig.11). Remarkably, both enzymes continued to display *trans*-cleavage activity up to 63 °C, and a low residual level of activity was still observed at 64 °C, beyond which the activity was completely abolished.

Together, these results suggest that amino acid substitutions alone are insufficient to improve the thermostability of LbCas12a, whereas optimizing the reaction buffer (e.g., using Tango buffer) can significantly enhance its high-temperature *trans*-cleavage activity.

The relevant experimental results and discussion have been incorporated into the revised manuscript (Page 6, Lines 179-190).

“Together, these results suggest that while crRNA engineering can improve high-temperature performance, intrinsic enzyme thermostability remains limiting. Thus, inspired by the observation that enAsCas12a retains its activity at elevated temperatures (60°C), we next examined whether the thermostability of LbCas12a could be improved by rational mutagenesis. To this end, we introduced three substitutions (D156R/G532R/K538R) in LbCas12a, analogous to the corresponding mutations (E174R/S542R/K548R) in enAsCas12a (Supplementary Fig. 9). However, the LbCas12a-Mut exhibited a temperature-dependent trans-cleavage pattern similar

to that of the *LbCas12a*-WT, showing no enhancement in activity above 57 °C (Supplementary Fig. 10a). In contrast, optimizing the reaction buffer (e.g., using Tango buffer) substantially enhanced high-temperature trans-cleavage activity (Supplementary Fig. 10b), allowing both *LbCas12a*-WT and *LbCas12a*-Mut to remain functional at temperatures where the standard buffer conditions failed (Supplementary Fig. 11).”

Supplementary Fig.9 | Structural comparison of enAsCas12a and *LbCas12a*-Mut. Predicted three-dimensional structural models of enAsCas12a (blue) and *LbCas12a*-Mut (green) were generated using AlphaFold3 (<https://alphafoldserver.com/>) and visualized with PyMOL (<https://www.pymol.org/>). Structural alignment revealed that the mutation sites—highlighted in red for enAsCas12a and in yellow for *LbCas12a*-Mut—are spatially well conserved and located within the same functional domain, suggesting that *LbCas12a*-Mut is likely to possess a similar functional mechanism to enAsCas12a.

Supplementary Fig.10 | Validation of the high-temperature trans-cleavage activity of *LbCas12a*-Mut under two buffer conditions. a Real-time fluorescence kinetics of four

crRNAs—including two heat-sensitive crRNAs (crRNA9 and crRNA17) and two heat-tolerant crRNAs (crRNA10 and crRNA20)—in LbCas12a buffer. **b** Real-time fluorescence kinetics of the same crRNAs in Tango buffer. The target used was the PCR-amplified S-gene fragment at a final concentration of 100 pM. Data are represented as mean \pm standard error (n=3 technical replicates).

Supplementary Fig.11 | Evaluation the upper temperature limit of trans-cleavage activity for two LbCas12a in Tango buffer. **a** Real-time fluorescence kinetics of LbCas12a-WT guided by two thermotolerant crRNAs (crRNA10 and crRNA20) in Tango buffer at temperatures from 60 to 65 °C. **b** Real-time fluorescence kinetics of LbCas12a-Mut guided by two thermotolerant crRNAs (crRNA10 and crRNA20) in Tango buffer at temperatures from 60 to 65 °C. The target used was the PCR-amplified S-gene fragment at a final concentration of 100 pM. Data are represented as mean \pm standard error (n=3 technical replicates).

Minor questions:

1. Please clarify the incubation conditions in Fig. 2, including whether any buffer was added during the pre-incubation/activation steps and whether Mg^{2+} was present (and at what concentration). Additionally, the Methods section should provide more detailed information on the experimental setup, including the exact composition of the Cas12a reaction buffer and the identity of ‘polyC’—for example, whether it refers to a defined 6-mer (C_6) or a different length.

Response:

We sincerely thank the reviewer for this valuable comment. In the experiments shown in Fig. 2d–g, the different sample addition sequences are indicated as steps ① and ② in the schematic diagrams, with the main components for each step labeled above the corresponding arrows. It should be noted that, in addition to the illustrated components, all step ① reactions also contained the reaction buffer and nuclease-free water.

The 1 \times reaction buffer consisted of 5 mM Tris-HCl (pH 9.0), 5 mM NaCl, 15 mM $MgCl_2$, and 0.01% CA-630. The “polyC” reporter refers to a single-stranded DNA probe composed of six cytosine bases (C_6) labeled with a FAM fluorophore and a BHQ1 quencher at the 5' and 3' ends, respectively.

In accordance with your suggestion, we have provided a detailed description of these experimental procedures and all component compositions in the revised

Methods section (Page 19, Lines 624–636).

“Evaluation of the thermal stability of LbCas12a reaction components

Thermal stability assays were carried out in a 20 μ L reaction system containing 1 \times Cas12a buffer (5 mM Tris-HCl, pH 9.0; 5 mM NaCl; 15 mM MgCl₂; 0.01% CA-630), 10 nM LbCas12a, 10 nM crRNA, 500 nM polyC6 reporter (FAM–6C–BHQ1), and 100 pM dsDNA targets. For each test, the component under evaluation (LbCas12a, crRNA, pre-assembled LbCas12a–crRNA RNP, or RNP pre-activated with target DNA) was pre-incubated in 1 \times Cas12a buffer supplemented with nuclease-free water at 37°C, 45°C, or 53°C for 30 min. After pre-incubation, the remaining components were added to complete the 20 μ L reaction, which was then incubated at 45°C for real-time fluorescence acquisition using the Bio-Rad CFX96 Touch Real-Time PCR System, with data points collected at one-minute intervals.”

2. To better demonstrate the advantages of the proposed detection method, the authors should present a comparison table with qPCR.

Response:

We sincerely thank the reviewer for this constructive suggestion. Following your advice, we have added a comparison table summarizing the key performance parameters of the POP-CRISPR assay and conventional qPCR, including reaction time, limit of detection, equipment requirements, and assay cost. This table has been incorporated into the revised Supplementary Materials to better highlight the advantages of our proposed detection method.

Table S12 Comparison between POP-CRISPR and qPCR

Parameter	POP-CRISPR	qPCR
Reaction mode	One-pot (RAA + Cas12a)	One-pot (thermal cycling)
Temperature	37°C (amplification) \rightarrow 45–55°C (CRISPR detection)	Cyclic heating
Time to result	Within 20 min	60-90 min
Analytical LOD	0.5aM (\approx 4-5 copies)	1-10 copies
Clinical sample sensitivity	HPV-16 Ct=35.7 MP Ct=36.2	Ct \leq 37 considered positive
Concordance	100% with qPCR	Gold standard
Mismatch discrimination	Single-base resolution (\geq 53°C, non-canonical PAMs)	Limited
Sample preparation	Rapid 2-min heat lysis for nucleic acid release	Standardized nucleic acid extraction required (>20 min)
Instrumentation	Portable isothermal fluorescence detector	Thermal cycler (bench-top)

3. The schematic in Fig. 6a is not sufficiently clear. It should be clarified whether the mini-device simultaneously performs both sample processing and detection

reactions, or if it is limited to temperature control. The specific workflow and underlying mechanism should be explicitly described. We recommend including a more intuitive process illustration in the figure.

Response:

We sincerely appreciate the reviewer's valuable suggestion. In response, we have revised Figure 6a to clearly illustrate the three major steps and their sequence in the POP-CRISPR workflow—sample collection, rapid nucleic acid release, and on-device detection. The figure legend has also been expanded to provide a more detailed description of the working process and underlying mechanism.

To further enhance clarity, we have included an additional schematic in the Supplementary Materials composed of real device photographs (Supplementary Fig.24), visually demonstrating the complete detection procedure using the miniaturized device.

It should be noted that the current version of the mini-device integrates a temperature control module, optical detection unit, power management system, and communication interface within a unified design. Given that sample lysis and detection require distinct temperature conditions, these two steps are currently conducted sequentially in different temperature zones on the same device. In future iterations, we plan to implement a multi-zone temperature control or fully automated system to further improve operational efficiency and portability.

Fig.6 On-site detection of MP using POP-CRISPR. **a** Workflow of POP-CRISPR for on-site testing for MP using mini-device. Green arrows and red numbers (①–③) indicate the three main steps and their sequential order in the detection workflow. First, nasopharyngeal swab clinical samples are collected and mixed with Chelex-100-containing lysis buffer, followed by rapid nucleic acid release on the mini-device at 80 °C for 2 min. The obtained nucleic acids are then added to the POP-CRISPR reaction system, which is performed on the mini-device using a poikilothermal one-pot protocol (37 °C for 10 min followed by 45 °C). The detection results can be interpreted via wireless connection to a smartphone application, serving as a reference for subsequent diagnosis and prognosis. **b** Scheme for standardized nucleic acid extraction and rapid lysis for nucleic acid release. **c** Evaluation the extraction efficiency of two nucleic acid extraction methods using POP-CRISPR. 27 clinical nasopharyngeal swab samples suspected of MP infection were tested. Values are shown as endpoint fluorescence at 60 minutes. The threshold value (red dashed line) was set based on the mean fluorescence of no target control plus three times the standard deviation. **d** Real-time fluorescence profiles obtained from POP-CRISPR experiments

performed on mini-device. The experimental procedure was as follows: 10 min at 37°C followed by 45°C, figure shows the real-time fluorescence signal of 45°C.

Rapid on-site diagnostic test within 20 minutes

Supplementary Fig.24 | The operation workflow of on-site MP detection by POP-CRISPR using mini-device.